



# GIR v1.0.0: a generalised impulse-response model for climate uncertainty and future scenario exploration

Nicholas J. Leach[1], Zebedee Nicholls[2,3], Stuart Jenkins[1], Christopher J. Smith[4], John Lynch[1], Michelle Cain[1], Bill Wu[1], Junichi Tsutsui[5], and Myles R. Allen[1,6]

[1]Department of Physics, Atmospheric Oceanic and Planetary Physics, University of Oxford, United Kingdom.
[2]Australian–German Climate and Energy College, University of Melbourne, Australia.
[3]School of Earth Sciences, University of Melbourne, Australia.
[4]School of Earth and Environment, University of Leeds, Leeds, UK.
[5]Environmental Science Laboratory, Central Research Institute of Electric Power Industry, Abiko-shi, Japan.
[6]Environmental Change Institute, University of Oxford, Oxford, UK.

**Correspondence:** Nicholas J. Leach (nicholas.leach@stx.ox.ac.uk)

**Abstract.** Here we present a Generalised Impulse Response (GIR) model for use in probabilistic future climate and scenario exploration, integrated assessment, policy analysis and teaching. This model is based on a set of only six equations, which correspond to the standard Impulse Response model used for greenhouse gas metric calculations by the IPCC, plus one physically-motivated additional equation to represent state-dependent feedbacks on the response timescales of each greenhouse

gas cycle. These six equations are simple and transparent enough to be easily understood and implemented in other models without reliance on the original source code, but flexible enough to reproduce observed well-mixed greenhouse gas (GHG) concentrations and atmospheric lifetimes, best-estimate effective radiative forcing, and temperature response. We describe the assumptions and methods used in selecting the default parameters, but emphasize that other methods would be equally valid: our focus here is on identifying a minimum level of structural complexity. The tunable nature of the model lends it to use as

a fully transparent emulator of complex Earth System Models, such as those participating in CMIP6, while also reproducing the behaviour of other simple climate models. We argue that this GIR model is adequate to reproduce the global temperature response to global emissions and effective radiative forcing, and that it should be used as a lowest-common denominator to provide consistency and continuity between different climate assessments. The model design is such that it can be written in tabular data analysis software, such as Excel, increasing the potential user base considerably.

## 1   Introduction

Earth System Models (ESMs) are vital tools for providing insight into the drivers behind Earth's climate system, as well as projecting impacts of future emissions. Large scale multi-model studies, such as the Coupled Model Intercomparison Projects (CMIPs), have been used in many reports to produce projections of what the future climate may look like based on a range of





different emissions scenarios and associated socio-economic narratives quantified by Integrated Assessment Models (IAMs). In addition to simulating both the past and possible future climates, these CMIPs extensively use theoretical experiments to try to constrain some of the key properties of the climate system, such as the equilibrium climate sensitivity [ECS, Collins et al. (2013)], or the transient climate response to cumulative carbon emissions [TCRE, Allen et al. (2009)].

While ESMs are integral to our current best understanding of the how the climate system responds to GHG emissions, and provide the current best predictions for what a future world might look like, they are so computationally expensive that we are only able to run a limited set of experiments during a CMIP. This constraint on the quantity of experiments and scenarios able to be simulated necessitates the use of simpler emulators to provide probabilistic assessments and explore additional experiments and scenarios. These emulators are often referred to as simple climate models (SCMs). In general, they are able

to simulate the globally averaged emission → concentration → radiative forcing → temperature response pathway, and can be tuned to emulate an individual ESM (or multi-model-mean). In terms of complexity, SCMs are considerably lighter than ESMs, both in terms of the runtime (most SCMs can run tens of thousands of years of simulation per minute on an "average" personal computer, whereas ESMs may take several hours to run a single year on hundreds of supercomputer processors), and the number of lines of code (SCMs tend to be composed on the order of thousands of lines, ESMs can be up to a million lines

(Alexander and Easterbrook, 2015)).

Several simple climate models are available, such as the two used in the Intergovernmental Panel on Climate Change (IPCC) Special Report on 1.5°C warming [SR15, IPCC (2018)]: FaIR v1.3 and MAGICC6. However, while these models are "simple" in comparison to the ESMs they emulate, they are often still not so simple as to allow new users to quickly and easily understand

the equations and processes behind their calculations let alone how to run them. This learning curve behind the use of these simple models reduces their usability, and has meant that different research groups tend to use the model they are most familiar with. This has led to a number of different simple climate models being used in single reports for identical tasks, reducing the overall consistency of the work and introducing an unnecessary additional level of complexity for non-specialists. We believe one key step towards a transparent and coherent process in IPCC Assessments would be the use of at least one common SCM

between working groups, allowing results to be directly comparable throughout. Such a model would be a "lowest common denominator" model within these reports (likely alongside domain specific models).

An important innovation of the IPCC 5th Assessment Report (Myhre et al., 2013) was the introduction of a fully transparent set of equations (the AR5-IR model) for use in the calculation of greenhouse gas metrics, but that model was not quite

adequate to reproduce the evolution of the integrated impulse response to emissions over time. The Finite amplitude Impulse Response (FaIR) model v1.0 (Millar et al., 2017) generalised the AR5-IR model for non-linearity in the carbon cycle. FaIR v1.0 uses four equations to model the atmospheric gas cycle and corresponding effective radiative forcing (ERF) impact of $CO_2$, and a further two (unchanged from the AR5-IRF) to emulate the climate system's thermal response to changes in ERF. Subsequently, Smith et al. (2017) added a representation of other greenhouse gases, significantly increasing the structural com-





plexity of the FaIR model in FaIR v1.3 while Tsutsui (2017) demonstrated that the inclusion of an additional thermal response timescale was necessary to reproduce the response to rapid forcing changes such as volcanic eruptions or idealised instantaneous $CO_2$-doubling experiments.

Here we show that all of these innovations can be captured in a single set of six equations outlined in figure 1. As such,
we have chosen to call the model GIR, for Generalised Impulse Response. Despite their simplicity, these equations are based on parameterisations of physical process such that the model can still be used to assess climate system properties such as the Equilibrium Climate Sensitivity (ECS) and Transient Climate Response (TCR). We describe the methods behind selecting a default parameter set and associated uncertainties for the model using some constraints from more complex models and existing literature, but ultimately chosen to accurately represent historical observations. We demonstrate that this parameter set
can closely replicate inferred properties of the climate system in both historical observations and more complex models. We compare GIR's response with other widely used simple climate models for a subset of the Shared Socioeconomic Pathways, [SSPs, Riahi et al. (2017)], all using their respective default parameter sets. Previous work (Joos et al., 2013; Tsutsui, 2017) indicates that the model parameters can be tuned to reproduce the global mean behaviour of ESMs, suggesting that GIR could act as an emulator for individual CMIP6 ESMs, something we aim to demonstrate when sufficient CMIP6 output is available.


GIR is sufficiently simple as to be able to be used in undergraduate and high-school teaching of climate change, and can illustrate some key properties of the climate system such as the warming impacts of different GHGs, or the implications of uncertainty in ECS and TCR. To allow students and other users unfamiliar with scientific programming languages (such as GIR's native language, Python) access to the model, we also provide a version of GIR written in Excel. We hope that this may open
exploration of the climate system to a large group of potential users who do not have the expertise to run presently-available SCMs. The simplicity of GIR additionally means that although we provide model code, users do not need to rely on this, and would be able to relatively quickly re-create it in whatever language they are familiar with, and whatever format fits their intended usage. This increases the potential of GIR for use in integrated assessment, as instead of having to adapt existing code to run within an existing architecture, new code for this model could be written in exactly the format and structure required.





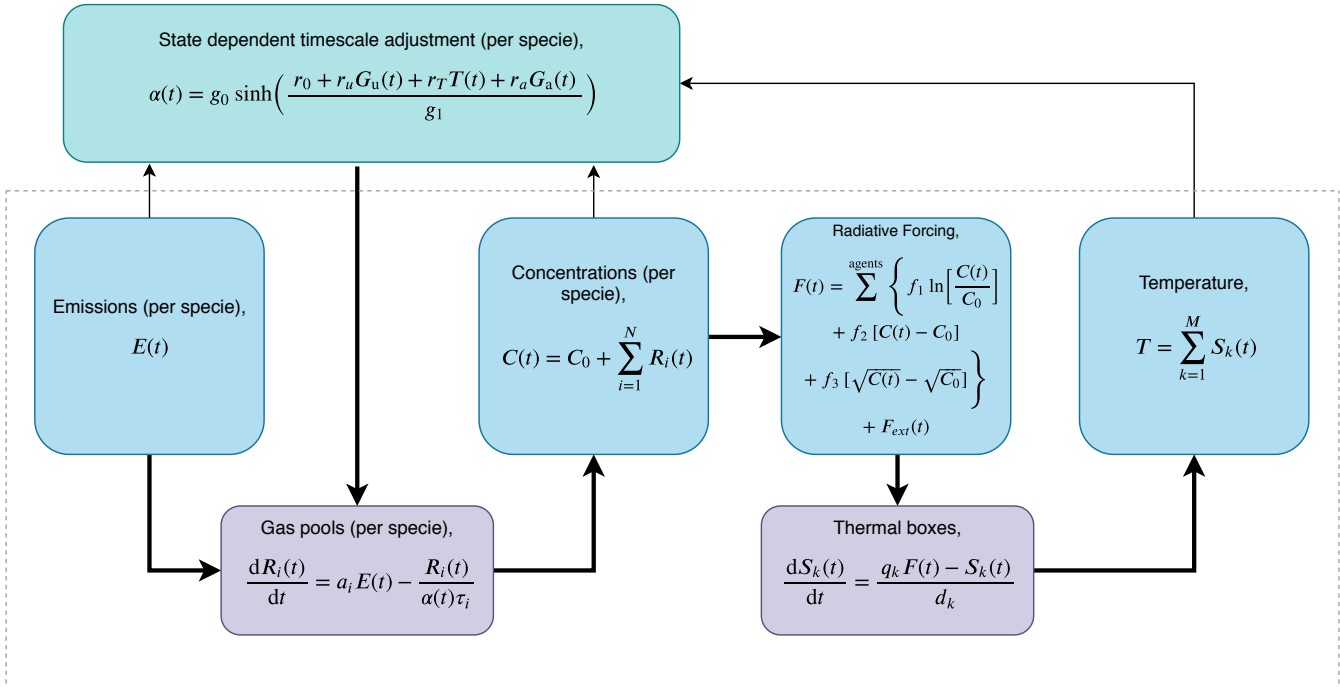

**Figure 1.** Schematic showing the full model structure and equations used. Model steps take place from left to right, with thick arrows indicating the flow of model steps that occur during timestep $t$, and thin arrows indicating steps that occur in between timesteps $t$ and $t + dt$. Equations are described in full below. The dashed grey line indicates the components identical to AR5-IR (Myhre et al., 2013).

## 2 Gas cycle model

GIR models the relationship between emissions and atmospheric concentrations of each gas species (the gas cycle) with three equations:

$$\frac{\mathrm{d}R_i(t)}{\mathrm{d}t} = a_i E(t) - \frac{R_i(t)}{\alpha \tau_i}, \tag{1}$$

$$C(t) = C_0 + \sum_{i=1}^{n} R_i(t) \text{ and} \tag{2}$$

$$\alpha(t) = g_0 \cdot \sinh\left(\frac{r_0 + r_u G_u(t) + r_T T(t) + r_a G_a(t)}{g_1}\right); \tag{3}$$

where $g_1 = \sum_{i=1}^{n} a_i \tau_i \left[1 - \left(1 + h/\tau_i\right)\mathrm{e}^{-h/\tau_i}\right]$

and $g_0 = \left[\sinh\left(\frac{\sum_{i=0}^{3} a_i \tau_i [1 - \mathrm{e}^{-h/\tau_i}]}{g_1}\right)\right]^{-1}$.

Equations 1 and 2 describe a gas cycle with an atmospheric burden above the pre-industrial concentration, $C_0$, formed of $n$ pools: each pool corresponds to a different sink from the atmosphere. Each pool, $R_i$, has an uptake fraction $a_i$ and decay





timescale $\tau_i$, which is multiplicatively adjusted by a state dependent factor, $\alpha$. At each timestep, $t$, $\alpha$ is computed, and then
the pool concentrations are updated and summed to determine the new atmospheric burden. $\alpha$ provides feedbacks to the gas
lifetimes based on the current timestep's levels of accumulated emissions ($G_u$), global temperature ($T$), and atmospheric gas
burden ($G_a$), with sensitivities encapsulated by the $r$ coefficients. The structural form of $\alpha$, (3), is an analytic approximation
to the numerical solution of the 100-year integrated Impulse Response Function (iIRF100) from equations 7 and 8 in Millar
et al. (2017). We include an additional term not in FaIR v1.0, dependent on $G_a$, the atmospheric mass burden, motivated by
the sensitivities of the CH$_4$ and N$_2$O lifetimes to their own abundances as predicted by atmospheric chemistry and simulated
in chemical transport models (CTMs) (Holmes et al., 2013; Prather et al., 2015). $g_0$ and $g_1$ set the value and gradient of our
analytic approximation equal to the numerical solution at $\alpha = 1$, but are not independent parameters.

We discuss the adequacy of this analytic form in the Supplementary Information and compare it to the numerical solution
used in FaIR v1.3 (Smith et al., 2017). While we do not include any lifetime feedbacks beyond those described above to avoid
having to specify additional exogenous variables and to maintain independence of the gas species, other feedbacks could the-
oretically be incorporated in the same framework by adding further $r$ coefficients to the model. We would, however, caution
against adding additional complexity unless it can be shown to be necessary to significantly improve the emulation of more
complex models in a policy-relevant application. For example, we have linearised the band-interaction terms in the calculation
of ERF due to CH$_4$ and N$_2$O: this approximation has some impact under very high concentration scenarios, but not under
ambitious or moderate mitigation scenarios that are the principal focus of policy.

**Observationally consistent parameter selection**

Here we discuss default parameter choices for the three greenhouse gases (GHGs) that have contributed the most to global
warming since pre-industrial times: carbon dioxide (CO$_2$), methane (CH$_4$) and nitrous oxide (N$_2$O). The CO$_2$ gas cycle is sim-
ilar to those in FaIR v1.0 (Millar et al., 2017) and FaIR v1.3 (Smith et al., 2017): a four carbon pool cycle with a decay timescale
dependence on $G_u$ and $T$. The CH$_4$ gas cycle has a single pool, with lifetime state dependencies on $T$ and $G_a$. While in reality,
CH$_4$ has several sinks (tropospheric OH, tropospheric Cl, stratospheric reactions and soil uptake) and a lifetime dependent on
many more variables (Holmes et al., 2013), we find that a single pool and the sensitivities given above are sufficient for repro-
ducing observed emission to concentration pathways. The N$_2$O gas cycle has a single pool and a lifetime dependent only on $G_a$.

For CO$_2$ we keep values of $a_i$ and $\tau_i$ identical to those in FaIR v1.0 and v1.3 (Millar et al., 2017; Smith et al., 2017), origi-
nally from AR5-IR (Myhre et al., 2013). $r_0$, $r_u$ and $r_T$ are tuned to reproduce present day concentrations – from the CMIP6
historical concentrations dataset, extended to 2017 with up-to-date measurements from NOAA (Meinshausen et al., 2017;
Battle et al., 1996; Butler et al., 1999) – when GIR is spun up from pre-industrial with bottom-up emission estimates from
the Global Carbon Project (Quéré et al., 2018) and run with a prescribed temperature pathway of attributable warming to the
present day [calculated as described in Haustein et al. (2017) using best-estimate forcings based on Forster et al. (2013) and the
mean of four temperature datasets (Vose et al., 2012; Cowtan and Way, 2014; Lenssen et al., 2019; Morice et al., 2011)] while





fixing the ratio of $r_u$ and $r_T$ to the FaIR v1.0 and v1.3 default. Uncertainties in $r_0$, $r_u$ and $r_T$ are taken from Millar et al. (2017).


The $CH_4$ gas cycle only has a single pool, so $a_i$ is zero except for $i = 1$. We set $\tau_1$ to the present-day lifetime of 9.15 years (Holmes et al., 2013). We fit $r_T$ and $r_a$ to the sensitivities of the $CH_4$ lifetime to tropospheric air temperature, tropospheric water vapour and $CH_4$ abundance in Holmes et al. (2013) by linearising GIR and the parametric model in Holmes et al. about 2010 values. We find that a global mean temperature dependence is sufficient to reproduce the sensitivities to both tropospheric air temperature and water vapour mixing ratio due to the close thermodynamic relationship between these quantities. We then fit the specified pre-industrial concentration and $r_0$ in an identical model run to the $CO_2$ parameter fitting procedure, but using bottom-up emission estimates from PRIMAP-histTP (Gütschow et al., 2016). While the parametric model in Holmes et al. includes lifetime sensitivities to eight other atmospheric variables such as anthropogenic NOx emissions, many of these have a small effect and all are scenario-dependent, requiring additional exogenous variables to be specified. Since we wish each gas cycle model to be independent we do not include any more state dependencies in addition to those described above. Uncertainties in $\tau_1$, $r_T$ and $r_a$ are derived from the uncertainties given in Holmes et al. (2013). We recommend that when using a Transient Climate Response (TCR) thermal parameter value that differs from the central estimate of 1.58K, both the $CO_2$ and $CH_4$ $r_T$ parameter values be scaled to match the ratio of the chosen TCR to the central estimate.

The $N_2O$ gas cycle only has a single pool, so $a_i$ is zero except for $i = 1$. We set $\tau_1$ to the present-day lifetime of 116 years (Prather et al., 2015). We fit $r_a$ to the sensitivity of the $N_2O$ lifetime to its own burden found in Prather et al. (2015) by linearising GIR and the parametric model in Prather et al. about 2010 values. We then fit the specified pre-industrial concentration and $r_0$ as for $CH_4$. All other variables that affect the $N_2O$ lifetime in Prather et al. are discarded to keep the gas cycle model independent as with $CH_4$. Uncertainties in $\tau_1$ and $r_a$ are derived from the uncertainties given in Prather et al. (2015).


The resulting tuned default parameters are given below, alongside their values in FaIR v1.0 and v1.3. Note that FaIR v1.3 has effective parameter values of $r_u, r_T, r_a = 0$ and a value of $r_0$ set such that $\alpha = 1$, and that FaIR v1.0 does not include $CH_4$ or $N_2O$. We emphasize that many choices have been made in selecting these parameters and different choices would lead to potentially quite different parameter values. One example of this is given in parentheses, in which we fit the $CH_4$ $r_0$ parameter by tuning inverse emissions from GIR to the best-estimate top-down anthropogenic emission estimate in the recent Global Methane Budget (Saunois et al., 2019), while setting the pre-industrial concentration to the 1500-1800 mean of 720 ppb from Meinshausen et al. (2017). Since Saunois et al. (2019) represents the most up-to-date assessments of global methane emissions, we choose this $CH_4$ $r_0$ and $C_0$ tuning as the default in GIR.

**Specification of natural emissions**

One key difference between GIR and FaIR v1.3 is the treatment of natural emissions of $CH_4$ and $N_2O$. FaIR v1.3 requires a quantity of natural emissions to be specified for these gases. Figure 2 in Smith et al. (2017) illustrates the time-dependent pathway of natural emissions required to reproduce the historical Representative Concentration Pathway (RCP) concentrations





**Table 1.** Default parameter values for the gas cycle components in different versions of FaIR. Where relevant, uncertainties for GIR are included as the 1-$\sigma$ range and are assumed to be normally distributed.

| Parameter | GIR | | | FaIR v1.3 | | | FaIR v1.0 | | |
|---|---|---|---|---|---|---|---|---|---|
| | $CO_2$ | $CH_4$ | $N_2O$ | $CO_2$ | $CH_4$ | $N_2O$ | $CO_2$ | $CH_4$ | $N_2O$ |
| $a_1$ | 0.2173 | 1 | 1 | 0.2173 | 1 | 1 | 0.2173 | - | - |
| $a_2$ | 0.2240 | 0 | 0 | 0.2240 | 0 | 0 | 0.2240 | - | - |
| $a_3$ | 0.2824 | 0 | 0 | 0.2824 | 0 | 0 | 0.2824 | - | - |
| $a_4$ | 0.2763 | 0 | 0 | 0.2763 | 0 | 0 | 0.2763 | - | - |
| $\tau_1$ | 1000000 | $9.15 \pm 10\%$ | $116 \pm 8\%$ | 1000000 | 9.3 | 121 | 1000000 | - | - |
| $\tau_2$ | 394.4 | - | - | 394.4 | - | - | 394.4 | - | - |
| $\tau_3$ | 36.54 | - | - | 36.54 | - | - | 36.54 | - | - |
| $\tau_4$ | 4.304 | - | - | 4.304 | - | - | 4.304 | - | - |
| $r_0$ | $28.63 \pm 8\%$ | 9.079 (8.445) | 67.84 | 35.0 | - | - | 32.4 | - | - |
| $r_u$ | $0.01977 \pm 8\%$ | 0 | 0 | 0.019 | - | - | 0.019 | - | - |
| $r_T$ | $4.334 \pm 8\%$ | $-0.2872 \pm 15\%$ | 0 | 4.165 | - | - | 4.165 | - | - |
| $r_a$ | 0 | $0.0003434 \pm 13\%$ | $-0.0009993 \pm 16\%$ | 0 | - | - | 0 | - | - |
| $C_0$ | 278 | 733.8 (720.0) | 271.3 | 278 | 722 | 273 | 278 | - | - |
| $E_2C$ | 0.4690 | 0.3517 | 0.2010 | 0.4690 | 0.3517 | 0.2010 | 0.4690 | - | - |

when the corresponding emission timeseries (Meinshausen et al., 2011b) are run through FaIR v1.3. These natural emission timeseries vary significantly over the recent past: $CH_4$ drops from 209Tg in 1765 to 139Tg in 1900 then rises to 191Tg in 2005,

at which it is fixed thereafter; and $N_2O$ remains relatively constant around 11Tg before dropping sharply to 9Tg around 1950; it is fixed at 8.99Tg beyond 2005. Another widely used simple climate model, MAGICC (Meinshausen et al., 2011a), does not do this explicitly, but is almost always run concentration driven until the present-day (an essentially identical procedure). This is required since the RCP database emission series were created in parallel to the concentration pathways used in CMIP5 using Integrated Assessment Models (Moss et al., 2010), resulting in inconsistent emission and concentration data. There is

little evidence for the high interdecadal trends of these natural emission pathways: Arora et al. (2018) finds the total change in natural $CH_4$ fluxes within CLASS-CTEM between the 1850s and 2000-2008 is +17Tg, and other studies assume that there have been no significant changes (Holmes et al., 2013; Prather et al., 2012). Due to the uncertainties associated with natural emissions of $CH_4$ and $N_2O$ (Turner et al., 2019; Davidson and Kanter, 2014), we have chosen to effectively fix our natural emissions by specifying a pre-industrial concentration over which anthropogenic emissions sources increase the atmospheric

concentration burden, identical to the $CO_2$ gas cycle in all FaIR versions, and tune our gas cycle parameters to bottom-up anthropogenic emissions (Gütschow et al., 2016; Quéré et al., 2018) and observed concentrations (Meinshausen et al., 2017). For comparison with FaIR v1.3, Figure 2 in the Supplementary Information shows the residual emissions required to reproduce





the RCP historical concentrations in GIR and the historical RCP emissions dataset. This is closely related to Figure 2 in Smith et al. (2017), indicating that while GIR and FaIR v1.3 take different approaches to natural emissions, the gas cycle models used

by each are similar.

We demonstrate below that we are able to reproduce observed historical concentrations from emissions without the need for specifying unphysical time-varying natural emissions. This avoids possible discontinuities at the present day caused by rapidly changing natural emissions or switching from concentration-driven to emission-driven modes of running SCMs. Avoiding

such discontinuities is particularly important for the analysis of ambitious mitigation scenarios, where the present-day state and trajectory of the climate system are key (Leach et al., 2018).

**Emission-driven historical simulations**

Here we run GIR with bottom-up emissions timeseries from the Global Carbon Project for $CO_2$ (Quéré et al., 2018) and PRIMAP-histTP (Gütschow et al., 2016) for $CH_4$ and $N_2O$. All other forcings are best-estimates from Forster et al. (2013).

Figure 2 shows the gas concentrations as simulated in GIR with default parameters and uncertainties as described in both the gas cycle and thermal response parameters, plotted alongside observed concentrations from the CMIP6 historical dataset (Meinshausen et al., 2017). Inset axes show the driving emission series used. We see that in general, GIR does well at matching the observed values, with the largest difference occurring for pre-1950 concentrations of $N_2O$. However, rather than this being a fundamental issue with the model, it is clear that the emission timeseries used is not compatible with the observed

concentrations, as while concentration observations start increasing pre-1900, emissions only begin to increase around 1940. This discrepancy has occurred in other $N_2O$ modelling studies using different methods and data sources (Saikawa et al., 2014). This is an issue of ongoing research (Tian et al., 2018), and it is possible that future work may provide a firm explanation for this discrepancy, allowing us to adjust GIR parameters accordingly. Crucially, the model does match the trend in observed concentrations over the recent period, where we would expect the emission estimates to be most accurate. We find that the pre-

industrial lifetimes of $CH_4$ and $N_2O$ are $9.05 \pm 0.4$ and $119.9 \pm 0.2$ years; pre-industrial natural emissions are $230 \pm 15 TgCH_4$ and $11.3 \pm 0.2 TgN_2O$-$N_2$; and present-day (2016) lifetimes are $10.1 \pm 0.9$ and $118.5 \pm 0.5$ years respectively. These quantities are within the assessed ranges found in more specific studies (Prather et al., 2012, 2015; Holmes et al., 2013; Kirschke et al., 2013; Davidson and Kanter, 2014; Arora et al., 2018). Note that the $CH_4$ values are calculated using the PRIMAP-hist tuned $CH_4$ $r_0$ value; if instead we use the value tuned to the Global Methane Budget, the $CH_4$ lifetimes are reduced by roughly 0.8

years.

**Other well-mixed GHGs**

While the focus of this study is the three major GHGs, GIR can also be used to simulate gas cycles for 40 other well-mixed halogenated GHGs found in the CMIP6 historical concentrations (Meinshausen et al., 2017). Figure 3 in the supplementary information shows GIR inverse emissions from these concentrations and inverse emission estimates from a more complex

atmospheric chemistry model (Cunnold et al., 1994; Rigby et al., 2011, 2014; Engel et al., 2018; Prinn et al., 2018). We note



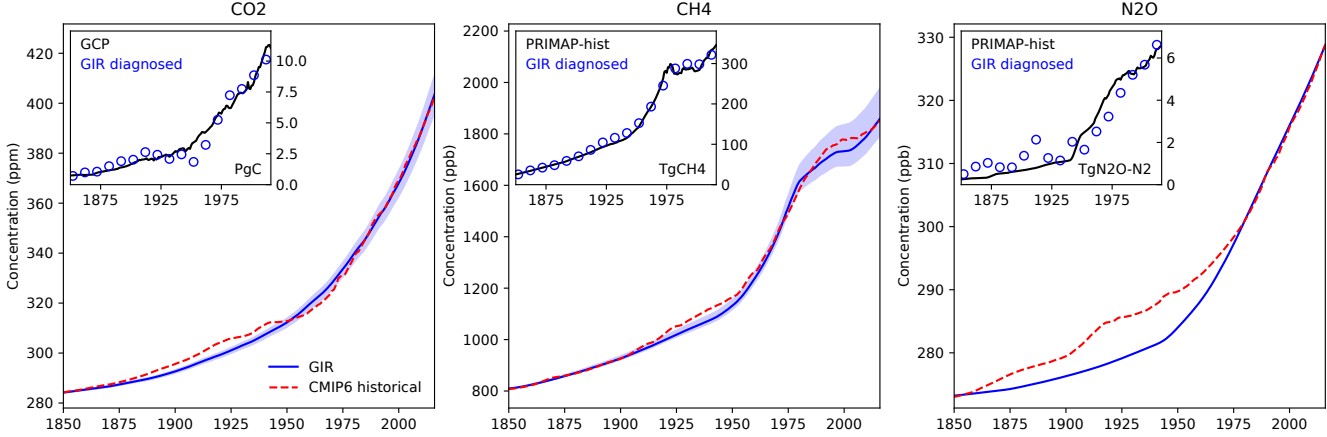

**Figure 2.** Historical simulations with GIR. Blue solid lines show GIR default concentration response to bottom-up emission data, and light blue plume shows 5-95% range. Dotted red lines show concentrations from Meinshausen et al.. Inset axes show corresponding bottom-up emission data, alongside decadal mean GIR inverse emissions diagnosed from Meinshausen et al. concentrations.

that for several gases – HFC-32, HFC-227ea, HFC-245fa, HFC-43-10-mee and C6F14 – GIR deviates significantly from the RCP database emissions. However, given the disagreement between observationally based estimates (Vollmer et al., 2011; Ivy et al., 2012; O'doherty et al., 2014) and the RCP emissions we do not regard this as being particularly problematic.

## 3 Concentrations to effective radiative forcing

GIR uses a simple formula to relate atmospheric gas concentrations to effective radiative forcing. This equation, below, includes logarithmic, square-root, and linear terms; motivated by the concentration-forcing relationships in Myhre et al. (2013) of $CO_2$, $CH_4$ and $N_2O$, and all other well-mixed GHGs respectively; however we allow each gas to be a more complex fit containing elements of each function if required. $F_{ext}$ is the sum of all forcings requiring to be specified exogenously. We suggest these may include natural forcing agents, aerosols (if the parameterisation below is not used), forcing due to albedo changes, and 215 forcing due to contrails.

$$F(t) = \sum^{\text{agents}} \left\{ f_1 \cdot \ln\left[\frac{C(t)}{C_0}\right] + f_2 \cdot [C(t) - C_0] + f_3 \cdot [\sqrt{C(t)} - \sqrt{C_0}] \right\} + F_{ext}, \tag{4}$$

where the $f$ coefficients are specified parameters, $C(t)$ is the concentration at time $t$ and $C_0$ is the pre-industrial concentration of the gas species.

**Selecting parameters based on Etminan et al. (2016)**

GIR is tuned by default to match the concentration-forcing relationships in Etminan et al. (2016). To maintain independence of the gases, in line with our core aim of simplicity, no interaction terms between gases are included. We first fit (by ordinary



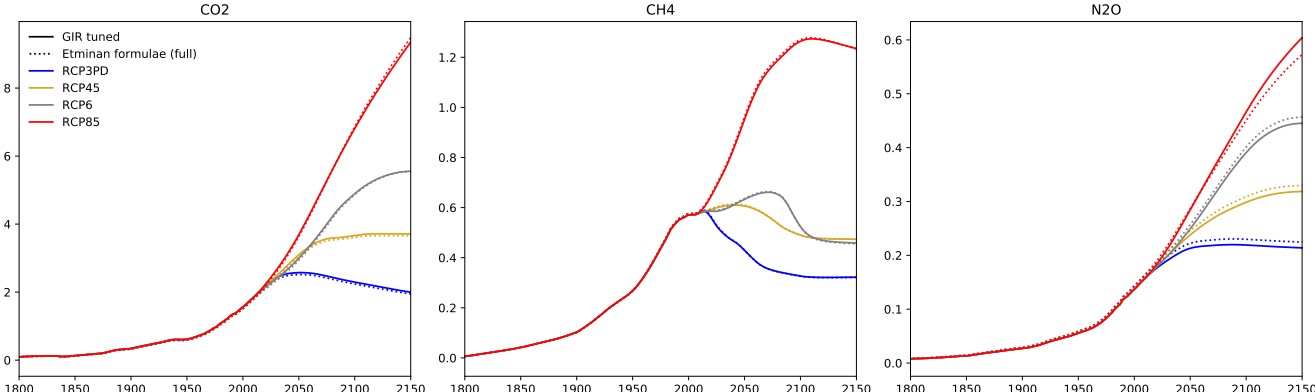

**Figure 3.** Comparison of the effective radiative forcings calculated in GIR versus the simple formulae derived from spectral measurements given in Etminan et al. (2016).

least-squares regression) the most physically relevant $f$ coefficient (logarithmic for $CO_2$ and square root for $N_2O$ and $CH_4$) to the Oslo line-by-line (OLBL) data from Etminan et al. (2016) Table S1, for each gas selecting data where the other two gases are at their present day concentrations and excluding data in which the gas is below pre-industrial levels (resulting in
three concentration/forcing data points per gas). This means that interactions between gases are included at their present-day levels. We then fit the other two $f$ parameters to the remaining residual. Fitting the most physically relevant parameter before the others ensures that the resulting fits extrapolate to higher concentrations without issue. Our fit parameters give an effective radiative forcing under a doubling of $CO_2$, $F_{2\times CO_2}$ of 3.84 $\text{Wm}^{-2}$. Figure 3 shows the radiative forcings of $CO_2$, $CH_4$ and $N_2O$ over the RCP database concentrations as computed using our equation in comparison to the simple formulae in Etminan
et al., both with and without interaction terms. We see that our equation very closely resembles the formulae in Etminan et al., only deviating slightly under the very high concentrations in RCP8.5. Figure 4 in the Supplementary Information shows the percentage error in the GIR equations compared to the simple formulae from Etminan et al.. These errors are less than the absolute error in the line-by-line calculations performed by Etminan et al.. However, we note here that the relatively larger error in $N_2O$ RF calculated for RCP8.5 is due to the very high $CH_4$ concentrations reached, arising from increasing coal production
throughout the 21st century, which has been suggested is unrealistic.

$CH_4$ concentrations also indirectly affect effective radiative forcing through oxidation to stratospheric water vapour and its effect on atmospheric ozone (Noël et al., 2018; Owens et al., 1982; Myhre et al., 2013). As well as the default parameters that do not include these indirect $CH_4$ forcings, we provide an alternative $CH_4$ $f_2$ parameter value including these indirect
forcings based on Myhre et al. (2013). The default parameters, and alternative $CH_4$ $f_2$ are given below, alongside present-day (2018) effective (direct) radiative forcings (ERF) and radiative efficiencies (RE) of each gas based on the CMIP6 historical concentration dataset (Meinshausen et al., 2017) and default GIR pre-industrial concentrations. We note that although the



**Table 2.** Default parameter values for the concentration-forcing relationships in GIR. An alternative $CH_4$ $f_2$ parameter value including indirect forcing effects; and corresponding present day metrics are given in parentheses.

| Parameter | $CO_2$ | $CH_4$ | $N_2O$ |
|---|---|---|---|
| $f_1$ | 5.754 | 0.06174 | -0.05441 |
| $f_2$ | 0.001215 | -0.000049 (0.000187) | 0.000157 |
| $f_3$ | -0.06960 | 0.03842 | 0.1062 |
| Present-day ERF ($Wm^{-2}$) | 2.11 | 0.621 (0.89) | 0.181 |
| Present-day RE ($Wm^{-2}ppm^{-1}$) | 0.0136 | 0.428 (0.664) | 2.91 |

$CH_4$ indirect effect does not account for the full radiative perturbation due to ozone, we can account for the full perturbation by setting the $CH_4$ $f_2$ parameter to 0.000339, as best-estimate total ozone forcing historically correlates highly with $CH_4$ concentrations. Hence we can avoid having to model ozone atmospheric chemistry, but still include its impacts on global climate through $CH_4$ parameterisation.

**Aerosol, other well-mixed greenhouse gas and all other forcings**

While the main focus of GIR is as a simple emulator to calculate climate impacts of $CO_2$, $CH_4$ and $N_2O$ emissions – inluding
other forcings exogenously – the equation set can be adapted to include forcing from aerosol emissions in a similar manner to existing simple models. As this is not the main focus of this study, here we outline the other studies from which we derive our aerosol parameters. Because of their short lifetimes, aerosol emissions are (through setting their lifetime to 1 year and their emission to concentration conversion factor to 1) converted directly to effective radiative forcings. In the default model version, linear emission to forcing ($f_2$) parameters for the direct aerosol radiative effect are taken from Aerocom II (Myhre
et al., 2013) for SOx, VOC, NH3, BC, OC and NOx emissions. As in Stevens (2015), effective radiative forcing from aerosol cloud interaction is assumed to be a logarithmic function of SOx emissions, which can be incorporated into GIR through setting the $C_0$ and $f_1$ parameters. By default we use a pre-industrial SOx emission value (analogous to $C_0$ in GIR terminology) of 60 $MtSO_2$ $yr^{-1}$, and set $f_1$ for $SO_x$ such that when CEDS emission data is run through GIR, the AR5 best-estimate aerosol cloud interaction ERF of -0.45 $Wm^{-2}$ is returned. All other well-mixed greenhouse gas effective radiative forcings are assumed to
be linear functions of their concentration ($f_1$ and $f_3 = 0$), with $f_2$ values taken as radiative efficiencies from Hodnebrog et al. (2013). All other forcings, such as that arising from albedo change or natural forcing changes are required to be inputted by the user. For a complete list of parameter values, follow the model introduction at https://github.com/njleach/GIR/blob/master/GIR/GIR_example_notebook.ipynb. To avoid any confusion over the default units required by each species in GIR, we provide a table listing the default units in the Supplementary Information.





## 4    The thermal response model

The thermal response model is a simple linear step response model (Good et al., 2011). While previous similar models have tended to use two thermal boxes (or analogously response timescales), in our GIR framework we leave the number of response timescales in GIR general, such that if more are desired, e.g. for a faster response to volcanoes or pulse experiments as in Tsutsui (2017), the user may specify how many to use. The two equations modelling the thermal response are below.

$$\frac{\mathrm{d}S_i(t)}{\mathrm{d}t} = \frac{q_j F(t) - S_i(t)}{d_i} \tag{5}$$

$$\text{and } T(t) = \sum_i^N S_i(t), \tag{6}$$

where $S_i$ represents the temperature anomaly in box $i$ with response timescale $d_i$ at time $t$, $F(t)$ is the effective radiative forcing, and $T(t)$ is the global mean surface temperature anomaly. As in Millar et al. (2017) and Tsutsui (2017), these equations can be solved analytically to relate $q_i$ and $d_i$ to the Transient Climate Response (TCR) and Equilibrium Climate Sensitivity (ECS) (Collins et al., 2013) as follows:

$$\text{ECS} = F_{2\times\text{CO}_2} \cdot \sum_i^N q_i \tag{7}$$

$$\text{TCR} = F_{2\times\text{CO}_2} \cdot \sum_i^N \left\{ q_i \left( 1 - \frac{d_i}{70} \left[ 1 - e^{-\frac{70}{d_i}} \right] \right) \right\}. \tag{8}$$

**Selected default parameters**

This thermal response model has been previously studied in detail (Geoffroy et al., 2013b, a; Gregory et al., 2015; Smith et al., 2017), and we therefore rely on existing studies to provide our default thermal response parameter distributions. While the 2-box response model has been more widely used in previous literature, we recommend that 3-boxes should be used due to the improvement seen in the emulation of more complex models (Tsutsui, 2017). Examining the 3-box CMIP6 model parameters from Tsutsui (2019), and ordering the response timescales longest to shortest (1-3), we sample parameters as follows. We use a gaussian distribution truncated at $\pm 2\sigma$ for $d_1$, with $[\mu, \sigma] = [283, 116]$; $d_2$, $d_3$ and $q_3$ are correlated and we sample these using a gaussian copula fit to the CMIP6 model parameters; the remaining two parameters, $q_1$ and $q_2$ are calculated from distributions of the TCR and realised warming fraction (TCR:ECS ratio, RWF) using equations 7 and 8. We specify a distribution for the RWF rather than the ECS since the RWF is shown to be more statistically independent of the TCR compared to the ECS (Millar et al., 2015). We assume a lognormally distributed TCR with 5-95 percentiles of 1.0 and 2.5 (Collins et al., 2013); and a normally distributed realised warming fraction (the TCR:ECS ratio, RWF) $N[0.58, 0.06]$, based on inferred CMIP6 RWF values (Tsutsui, 2019). The parameter distributions described above are used to generate a range in the thermal response throughout this study. While we focus on the use of GIR as a "stochastic" model to provide a range of possible responses corresponding to likely parameter distributions, the median thermal response parameters are given below.





**Table 3.** Median parameter values for the thermal response model in GIR.

| Parameter | $d_1$ | $d_2$ | $d_3$ | $q_1$ | $q_2$ | $q_3$ | ECS | TCR |
|---|---|---|---|---|---|---|---|---|
| 3-box | 283 | 9.88 | 0.85 | 0.328 | 0.175 | 0.242 | 2.85 | 1.63 |
| 2-box | 218 | 4.15 | - | 0.325 | 0.392 | - | 2.73 | 1.58 |

**Parameter choice and the thermal response of simple models**

One point of contention within the recent Special Report on 1.5°C warming (IPCC, 2018) was the difference in apparent
response between the two simple models used in the report: FaIR v1.3 (henceforth FaIR) and MAGICC6 (MAGICC). The
FaIR thermal response lies significantly below the MAGICC response in the published model outputs from the IAMC scenarios
(Huppmann et al., 2018). Figures 5 and 6 below include the median total forcings and temperature responses in each model for
three Shared Socioeconomic Pathways (Riahi et al., 2017). This shows that while the total forcings derived from emissions by
each model are relatively similar, with FaIR v1.3 simulating a slightly wider inter-scenario range of forcings than MAGICC, the
temperature response in both FaIR v1.3 and GIR is consistently lower than MAGICC, particularly notable for the high emission
SSP5-Baseline scenario. This difference in response is likely due to the default parameter choice in each model, with MAGICC
response parameters tuned to output from CMIP3 atmosphere and oceanic general circulation models (AOGCMs) and AR4
best estimates of aerosol radiative forcing (Meinshausen et al., 2011a), while the response in FaIR v1.3 (a 2-box variant of the
GIR thermal model) was tuned to constrain modelled response to the observed temperature changes from Cowtan and Way
(2014) (Smith et al., 2017). Another reason for divergent temperature projections is that FaIR used newer AR5 distributions
for ECS and TCR, which are lower than the posterior distribution from Meinshausen et al. (2009), as well as the less negative
aerosol radiative forcing, resulting in lower temperature projections. Here we emphasize that the models themselves are not
systematically biased either low or high — it is the parameters used, and how these are selected, that determines the model
response. In GIR we have chosen to use the AR5 synthesis values for our thermal response parameters as above, which will
have a lower default response than both MAGICC and the CMIP3, CMIP5 and CMIP6 AOGCM means, but any user can tune
parameters based on what they aim for GIR to emulate. For example, we also provide GIR parameters (3-box thermal and
$CO_2$ radiative forcing parameters) tuned to a set of CMIP6 models (Tsutsui, 2019). These parameters correspond to a thermal
response range more similar to MAGICC6 than the default as seen below in Figure 6, illustrating that the choice of model
parameters is more important than the choice of model itself (at least for SCMs).

**5   Idealised and scenario experiments in GIR**

In this section we run several standard "benchmark" tests that have been carried out by previous simple modelling groups.
These include simple pulse experiments such as in Joos et al. (2013), and comparing simulations of the SSPs (Riahi et al.,
2017) from other simple models used in SR15 (Huppmann et al., 2018; IPCC, 2018) with that from GIR. We also run the





standard RCP concentrations backwards and the corresponding but not fully compatible due to their parallel development
(Moss et al., 2010) emission scenarios forwards through GIR in the Supplementary Information.

**Idealised Experiments**

First, we run a series of standard pulse experiments. GIR is spun up to present-day (2019) with inverse emissions derived from
historical concentrations (Meinshausen et al., 2017), then concentrations of $CO_2$, $CH_4$ and $N_2O$ are fixed at 407.9 ppm, 1867
ppb and 330.8 ppb respectively; updating and approximately replicating the experimental protocol of Joos et al. (2013). All
other forcings are fixed at the 2019 value. One run is carried out with no additional emissions, and the other three have an
emission pulse of 1Tg of one of the gases added in 2019. A 1Tg pulse is used since unlike in the majority of models in Joos
et al. (2013), GIR has no internal variability, so we do not require a pulse as large as 100Pg to obtain sufficient signal; regard-
less, the results obtained are independent of the pulse size over several orders of magnitude (for $N_2O$ and $CH_4$ at least until
the equations begin to break down due to, for example, lifetime feedbacks causing unphysical negative lifetimes). We derive
four metrics for each gas: the instantaneous airborne fraction after 100 years ($IRF_{100}$) – the proportion of gas from the initial
pulse remaining in the atmosphere; the integrated Impulse Response Function after 100 years ($iIRF_{100}$) – the integral of the
instantaneous airborne fraction over the 100 years following the initial pulse; the Absolute Global Warming Potential (AGWP)
– the integrated radiative forcing over a specified number of years following the pulse, normalized per kg of gas; the Global
Warming Potential (GWP) after 20 and 100 years – a very widely used metric of the potency of a GHG equal to the AGWP of
the GHG divided by the AGWP of $CO_2$ over the given time period; and the initial pulse-adjustment timescale (IPT) – the time
between the initial pulse and peak warming. Both $CH_4$ and $N_2O$ also contribute indirectly to radiative forcing through their
atmospheric chemistry, hence GWP values are given for direct-only forcings and full forcings, for $N_2O$ taking the absolute
differences as calculated in Myhre et al. (2013), and for $CH_4$ using the alternative $CH_4$ $f_2$ parameter including indirect forc-
ings. We find values comparable to the current literature (Joos et al., 2013; Ricke and Caldeira, 2014; Collins et al., 2019). Our
central calculated IPT value for $CO_2$ agrees well with Zickfeld and Herrington (2015); Ricke and Caldeira (2014), however we
find that while the IPT does increase with the pulse size, it does not to the extent in Zickfeld and Herrington; only by 2 – 5 years
for a pulse of 1000PgC. However, we note that unlike a 2-box thermal model (Ricke and Caldeira, 2014), the 3-box model
does not bound the IPT significantly; we believe that this metric could therefore be used to further constrain the 3-box thermal
parameter distributions in future work. This will require ensemble simulations with ESMs, possibly supported by observed
responses to volcanic eruptions. A related, but distinct metric that may be used to constrain both the 3-box thermal parameters
and IPT based on ESM ensembles is the initial rate of warming in response to a step change in forcing – the Step Response
Timescale (SRT). This is likely more suitable since the IPT is potentially an unstable definition of the time to peak warming
(since the IPT can be vastly different even if two responses realise *nearly* all their warming in the same time). It is possible that
a better definition of the IPT would be the time to realise 90% of the peak warming. While most metrics given below do not
change significantly if calculated with a more "real-world" baseline scenario, the $iIRF_{100}$ and instantaneous airborne fraction
for $CO_2$ do, thus we also give values for these metrics under an RCP4.5 baseline (the diagnosed inverse emissions that return

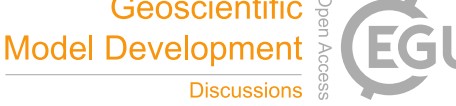

**Table 4.** Gas metric values from present-day pulse experiments. Uncertainties are provided as 5-95% ranges.

| Metric | $CO_2$ | $CH_4$ | $N_2O$ |
|---|---|---|---|
| $iIRF_{100}$ (yr) | 49.2 [44.9 , 54.1] | 10.8 [9.5 , 12.7] | $67.3 \pm 0.15$ |
| $iIRF_{100}$ (yr) – RCP4.5 | 56.6 [51.3 , 63.1] | 10.2 [9.15 , 11.8] | $67.5 \pm 0.2$ |
| $IRF_{100}$ (%) | 42.2 [38.4 , 46.5] | $0.0 \pm 0.0$ | $42.5 \pm 0.25$ |
| $IRF_{100}$ (%) – RCP4.5 | 52.9 [47.2 , 60.5] | $0.0 \pm 0.0$ | $41.9 \pm 0.4$ |
| $AGWP_{100}$ ($10^{-13}$ W m$^{-2}$ yr kg$^{-1}$) | 0.852 [0.7799 , 0.914] | 16.3 [14.8 , 18.5] | $250 \pm 0.5$ |
| $GWP_{100}$ (direct-only) | $1.0 \pm 0.0$ | 19.1 [16.8 , 22.3] | 294 [274 , 314] |
| $GWP_{100}$ (full) | $1.0 \pm 0.0$ | 30.2 [25.2 , 35.3] | 273 [253 , 293] |
| $AGWP_{20}$ ($10^{-13}$ W m$^{-2}$ yr kg$^{-1}$) | 0.221 [0.209 , 0.231] | 13.8 [13.0 , 14.7] | $68.7 \pm 0.0$ |
| $GWP_{20}$ (direct-only) | $1.0 \pm 0.0$ | 62.5 [57.7 , 68.5] | 312 [297 , 330] |
| $GWP_{20}$ (full) | $1.0 \pm 0.0$ | 96.7 [87.8 , 106] | 291 [276 , 309] |
| IPT (years) | 11.0 [2.0 , >100] | 3.5 [2.0 , 6.0] | 20.1 [7.1 , 32.0] |

the RCP database concentrations when run through GIR).

We also diagnose the Transient Response to Cumulative Carbon Emissions (TCRE) within GIR. This is the emergent linear
relationship between global temperature anomalies and cumulative $CO_2$ emissions, attributed to a balance between the increasing airborne $CO_2$ fraction as atmospheric concentrations are raised and the approximately logarithmic concentration-forcing relationship for $CO_2$. Here it is calculated in two ways: once following Smith et al., in which the temperature anomaly at 1000PgC was calculated when RCP8.5 was simulated by FaIR v1.3 (here we carry out this method using RCP4.5, 6 and 8.5, running GIR with concentrations and diagnosing both inverse $CO_2$ emissions and temperature anomalies). This method allows
both the $CO_2$-only TCRE and effective-TCRE (when all forcing agents are included in the temperature response) to be determined. The effective-TCRE is not a strict relationship (unlike the $CO_2$-only TCRE), and depends considerably on the non-$CO_2$ forcing pathway taken. For this reason, the effective-TCRE varies considerably between RCPs. The second method used to diagnose the TCRE ($CO_2$-only) follows Gillett et al. (2013), in which we prescribe a 1% per year increase in $CO_2$ concentrations starting from pre-industrial levels, and determine the ratio of global-mean warming to cumulative $CO_2$ emissions at the point of
$CO_2$ concentration doubling. The results are summarised in table 5 below. The result using the RCP 8.5 scenario is very similar to FaIR v1.3, though with a marginally higher median value, and is within the AR5 assessed range (Collins et al., 2013). The 1% experiment result of 1.45 [0.91 , 2.21] K (EgC)$^{-1}$ is consistent with a recent study (Millar and Friedlingstein, 2018) that utilised both the historic record and ESMs, though our result lowers the upper end of the TCRE distribution and raises the median. GIR is consistent with the best-estimate effective TCRE in Millar and Friedlingstein. A third method for calculating the
TCRE is to convert total anthropogenic forcing into $CO_2$-forcing-equivalent emissions (Jenkins et al., 2018), and regress this against corresponding attributable wwarming (Haustein et al., 2017). While we have not carried out this calculation here, one





**Table 5.** Diagnosed TCRE values in GIR.

| Method | Result [K (EgC)$^{-1}$] |
|---|---|
| **Effective-TCRE** | |
| RCP8.5 | 1.91 [1.20 , 2.90] |
| RCP6 | 1.76 [1.11 , 2.71] |
| RCP4.5 | 1.81 [1.14 , 2.79] |
| **CO$_2$-only TCRE** | |
| RCP8.5 | 1.46 [0.94 , 2.19] |
| RCP6 | 1.46 [0.93 , 2.22] |
| RCP4.5 | 1.46 [0.93 , 2.22] |
| 1% experiment | 1.45 [0.91 , 2.21] |

potential application of GIR would be the calculation of $CO_2$-forcing-equivalent emissions (and other diagnosed emissions), due to the ease of inversion of the gas cycles.

**Simulations of the Shared Socioeconomic Pathways**

Although we include RCP simulations with GIR in the Supplementary Information to allow comparison with the majority of the literature used, they have now been superseded by the SSP scenarios used in SR15 (Riahi et al., 2017; IPCC, 2018). Here we explore emission-driven simulations of three of the SSPs (each using the assigned integrated assessment model): SSP1-19, SSP2-45 and SSP5-Baseline. These three pathways span the range of possible future emissions and corresponding climates. Since the emissions series provided in the IAMC 1.5C database (Huppmann et al., 2018) only begin in 2005, we must har-

monize these to historical series to spin up GIR. We do this by diagnosing historical emissions for $CO_2$, $CH_4$ and $N_2O$ from the CMIP6 historical concentrations (Meinshausen et al., 2017) between 1750 and 2015 in GIR, then joining these to each SSP scenario in 2015, multiplying the SSP scenario by a linearly time dependent scaling factor – such that the SSP scenario emissions are equal to the diagnosed emissions in 2015 – that reaches a value of 1 by 2025. All other SSP forcings are harmonized identically, using best-estimate historical forcings from Forster et al. (2013). While this harmonization procedure may

not exactly match the one used in the database scenarios, present-day $CO_2$ concentrations and total forcing is similar, and this should therefore be a representative comparison (though we hope that emissions projections will also publish their historical estimates in future to avoid the need for re-harmonisation by other users). The GIR simulated concentrations are shown in figure 4. Concentration data from other SCMs (FaIR v1.3 and MAGICC6) is only publicly available for $CO_2$. We see that GIR simulates slightly lower $CO_2$ concentrations than the other SCMs, but that they generally lie within GIR 5-95% ranges. The

small $N_2O$ spread is due to its long lifetime and single atmospheric sink as modelled in GIR.

Median radiative forcing for each gas is available for both FaIR v1.3 and MAGICC6 in the IAMC database, so here we may



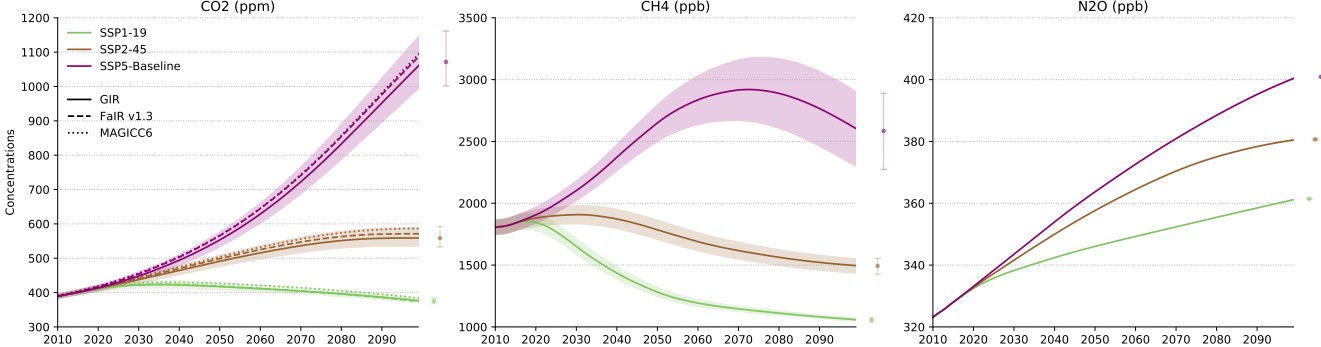

**Figure 4.** $CO_2$, $CH_4$ and $N_2O$ concentrations for a range of SSP scenarios. Solid lines show concentrations simulated in GIR from harmonized emissions data as described above. Dashed lines show $CO_2$ concentrations in FaIR v1.3; and dotted lines show concentrations in MAGICC6, from Huppmann et al. (2018).

do a more complete comparison between the three SCMs. Figure 5 shows that GIR has very similar $CO_2$ forcing to the other SCMs over the 21st century, lying between them for SSP5-Baseline and SSP1-19, and slightly below for SSP2-45. The other two SCMs lie within the GIR 5-95% ranges for all time. The $CH_4$ forcing highlights the difference in the $CH_4$ concentration-

forcing relationship between the models. Both GIR and FaIR v1.3 are based on the more recent study by Etminan et al. (2016), which revised present-day $CH_4$ forcing upwards by 25%, while MAGICC6 is based on Myhre et al. (2013). The difference between GIR and FaIR v1.3 is largely due to the different gas cycles: GIR has an interactive $CH_4$ lifetime with a positive self-abundance feedback and negative temperature feedback, while FaIR v1.3 keeps the $CH_4$ lifetime constant at 9.3 years throughout (Smith et al., 2017). GIR displays more similar $N_2O$ forcing to MAGICC6, while FaIR v1.3 lies somewhat below.

This discrepancy lies with the $N_2O$ gas cycle in FaIR v1.3, since the concentration-forcing relations in FaIR v1.3 and v2.0 are extremely similar, and is likely due to its relatively low estimate of natural $N_2O$ emissions at the present-day of 9.1 $TgN_2O-N_2$ (the value at which future natural emissions are fixed at). This value is taken from Prather et al. (2012), but a constant lifetime of 121 years is used, rather than the present-day lifetime estimate from that study of 131 years. This combination of low lifetime and natural emission estimate is likely the cause of the lowered future concentrations and corresponding forcing. The evolution

of total forcing in each scenario is highly comparable between the models, with GIR lying marginally above the other two for the SSP5-Baseline scenario – due to its higher $CH_4$ forcing – and in between for the two lower emission scenarios.

Finally, we compare the probabilistic temperature anomaly in each model. Figure 6 shows the median and 5-95% temperature response (min/max for CMIP6 GIR) in default GIR, FaIR v1.3, MAGICC6, and GIR tuned to 22 CMIP6 models (3-timescale thermal response, parameters following Tsutsui (2019)). Both GIR responses are relative to an anomaly of 0.61K in 1986:2005

(Kirtman et al., 2013), and the FaIR v1.3 and MAGICC6 responses are taken directly from the database (median 2010 anomalies are 0.93, 0.86, 0.89 and 1.09 in GIR, FaIR v1.3, MAGICC6 and CMIP6 GIR respectively). We see that the default GIR response is very similar, if marginally higher than the response of FaIR v1.3. Both FaIR v1.3 and GIR under default parameters have a significantly lower median response than MAGICC6, as well as a lower 5-95% spread for all scenarios. The CMIP6-





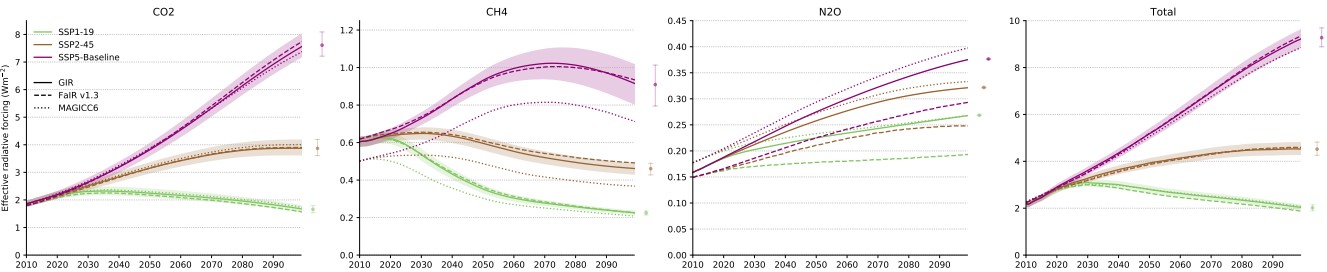

**Figure 5.** $CO_2$, $CH_4$ and $N_2O$ effective radiative forcing for a range of SSP scenarios. Solid lines show ERF simulated in GIR from harmonized emissions data as described above. Dashed lines show ERF in FaIR v1.3; and dotted lines show concentrations in MAGICC6, from Huppmann et al. (2018).

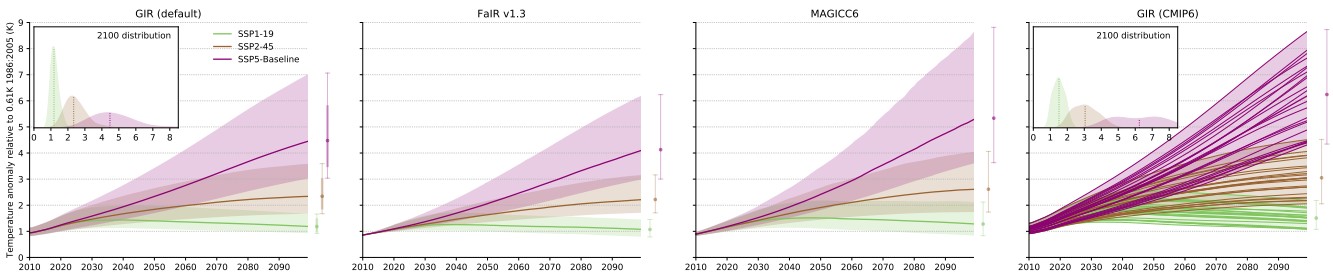

**Figure 6.** Temperature anomaly projections for a range of SSPs. Solid lines and plumes show median and 5-95% ranges (min/max for GIR CMIP6 emulation) for each model. Thin lines in GIR CMIP6 emulation plumes show emulations for individual CMIP6 models. Inset axes for GIR simulations show distribution of temperatures in 2100.

tuned GIR has a higher median response and lower spread than MAGICC6 for all scenarios, demonstrating the importance of
thermal parameter choice in SCMs, since this is fundamentally the same model as default GIR, but simulates a very different response.

## 6  Use of GIR

We envisage that GIR will primarily be used for similar assessments as are carried out with the current SCMs, such as providing probabilistic projections of atmospheric concentrations, radiative forcings and temperature anomalies for wide ranges
of scenarios, such as was carried out to provide the SR15 scenario explorer (Huppmann et al., 2018). GIR could also easily be coupled to integrated assessment models (IAMs) to explore impacts of climate policy options. One advantage that GIR has over the other SCMs discussed here, MAGICC6 (which is already used in several IAMs) and FaIR v1.3, is its relative computational efficiency and simple equation set. Due to its independent treatment of all the gas species and analytic approximations, it can be parallelised in a programming language designed for array operations (such as Fortran, MATLAB, or the NumPy Python mod-
ule) and hence GIR is extremely quick to run. For example, using its present Python implementation, GIR can compute around



2 million years per second for a three gas emission → concentration → radiative forcing → temperature pathway [1]. This speed provides significant advantages when computing large probabilistic ensembles, or when optimizing parameters dependent on the climate system. An important consideration for users computing probabilistic ensembles will be the memory required by GIR output, as this is significantly more likely to be the limiting factor on a modern computer, rather than the model runtime.


The six simple equations used by GIR also provide advantages for integration into IAMs, as rather than having to integrate the original SCM code into an existing architecture, the GIR equations set could be written up using a language and format consistent with the IAM codebase with limited effort. Their simplicity means that GIR can be run in programs for analysis of tabular data, including (but not limited to) Excel. This opens up climate system exploration to a large group of potential

new users who are familiar with spreadsheets, but not formal scientific programming languages. Finally, their simplicity adds considerably to the overall transparency of GIR; if it displays unexpected behaviour, it should be easy to determine where exactly this behaviour originates and explain it.

We suggest that the speed, simplicity and transparency of GIR lends it to use in undergraduate and high-school education

in addition to scientific research. It can be used to explain (and demonstrate) various aspects of both the carbon (or methane / nitrous oxide) cycle and Earth's thermal response to radiative forcing, and is simple enough to use that students could themselves carry out experiments (such as a $CO_2$ doubling) easily without prior experience and only basic computing skills.

GIR can also be used to rapidly investigate differences between ESMs, tuning GIR to emulate different full models and

comparing differences between the tuned parameter sets to identify which aspects of the models differ most, as was done with MAGICC in Meinshausen et al. (2011a). While CMIP6 emulation thermal response parameters are already available (Tsutsui, 2019), in order to fully emulate the CMIP6 ensemble, we will need to tune GIR to the gas cycles in individual ESMs as well, such that the range of carbon cycles responses is simulated. The ability to tune GIR (Tsutsui, 2017; Joos et al., 2013; Millar et al., 2017) to more complex models also allows estimation of complex model response to a particular scenario or experi-

ment without having to expend computer power to run the model itself; which could allow climate system uncertainties to be introduced more fully into integrated assessment studies by emulating the full CMIP6 ensemble within IAMs (providing the same capability as demonstrated by Meinshausen et al. (2011a) with an even faster model – though this remains to be properly demonstrated).


---

[1] on a laptop with 31GB RAM and an Intel(R) Core(TM) i7-8750H@2.2GHz



## 7   Conclusions

In this paper we have presented a simple gas-cycle impulse-response model. This represents a significant modification to the FaIR v1.0 SCM, removing the requirement for a non-linear equation to be numerically solved thanks to an analytic approximation, and adding a term in the state-dependence that depends on the atmospheric mass burden of the gas hence allowing

$CH_4$ and $N_2O$ to be modelled by the same equations as $CO_2$. This state-dependence allows important properties of $CH_4$ and $N_2O$ atmospheric chemistry to be included. Inclusion of these physically justified feedbacks should provide a more accurate emission-concentration relationship, particularly for scenarios in which feedbacks are significant such as high-emission scenarios, than models using a constant lifetime. The default gas-cycle parameters are tuned based on best-estimate observations of atmospheric concentrations and bottom-up emission estimates (Meinshausen et al., 2017; Gütschow et al., 2016; Saunois

et al., 2019), or are derived from literature (Prather et al., 2015; Holmes et al., 2013; Joos et al., 2013). We use a generalised equation to model the concentration-forcing relationships of $CO_2$, $CH_4$ and $N_2O$, demonstrating that with the calculated default parameters we can reproduce the equations derived from spectral measurements in Etminan et al. (2016). However, as it significantly simplifies the interpretation of model output for policy, we fix the interaction terms in the Etminan et al. (2016) equations at the present-day value. This does not affect results significantly except for scenarios with extreme-high emissions

of $CH_4$, such as RCP8.5. The thermal response model is identical to that investigated in many previous studies (Millar et al., 2017; Smith et al., 2017; Good et al., 2011; Geoffroy et al., 2013b, a; Tsutsui, 2017), so we do not focus on this component of GIR. However, we do recommend that users of GIR take care when tuning or selecting the parameters used in the thermal response component, as this has been a historically contentious issue when comparing the results of different SCMs. As such, we emphasize that GIR itself (as with most other SCMs) has neither a high nor a low response, since this is dependent entirely

on the parameters chosen. We give default parameters consistent with the assessed quantities of ECS and TCR in AR5 (Myhre et al., 2013), but other parameter choices would be equally valid.

We find that we are able to reproduce atmospheric concentrations from bottom-up emission estimates accurately, without the need for specifying time-dependent natural emissions (which have very high associated uncertainties). Some discrepancies between observed concentrations and emissions estimates cannot be explained by GIR, most notably pre-1980 $N_2O$ concen-

trations, but this is due to the chosen emission estimate, and future research may reconcile the two. Calculations of standard gas-cycle model comparison metrics agree well with values from the literature using methods following Joos et al. (2013). Most notably, we find that the GWP$_{100}$ of $CH_4$ is increased to 35.9 [31.2 , 42.5], due to compounding increases in both its lifetime and radiative forcing impact (Etminan et al., 2016; Holmes et al., 2013). We find policy significant differences in the

timing of peak warming following pulses of $CO_2$, $CH_4$ and $N_2O$: a pulse of $N_2O$ will cause a peak in attributed warming approximately 10 years after a pulse of $CO_2$ of equal size. We find that the GIR gas cycle, combined with AR5 estimates of ECS and TCR (Myhre et al., 2013) (assuming log-normal distributions), implies a TCRE of 1.45 [0.91 , 2.21] K (EgC)$^{-1}$, very similar to previous estimates (Millar and Friedlingstein, 2018). In order to more robustly assess relevant uncertainties in the carbon cycle, we will need to repeat this experiment with gas cycle parameters which emulate the CMIP6 ESM responses.






A comparison of simulations of selected SSPs within GIR against other widely used SCMs, FaIR v1.3 and MAGICC6, high-lights differences between the three models. These can almost entirely be explained by parameter selection in each model, and is caused by differences in the studies or data sources which each model has been tuned to reproduce; such as the concentration-forcing relationships in MAGICC6 following the simple relations in Myhre et al. (2013) while in FaIR v1.3 and this model they

follow the updated relations in Etminan et al. (2016). Crucially, we demonstrate that differences between the thermal response in these models is almost entirely dependent on the parameter selection, demonstrating this using two different parameter sets within our single model: one tuned to individual GCMs in the CMIP6 ensemble (Tsutsui, 2017), and one using the assessed range in AR5 (Collins et al., 2013). All three models are contributing to RCMIP Nicholls et al., in which their differences will be fully explored.


There are many potential uses for GIR as a result of its simplicity and transparency. In addition to being used for the same probabilistic scenario assessment as is carried out by SCMs in reports such as SR15 (IPCC, 2018), it could be very easily implemented into IAMs; and may provide some improvements in terms of computational efficiency due to its extremely rapid runtime. We also encourage GIR to be used by policy-makers in order to directly assess whether warming implications are

aligned with the intended outcomes of mitigation policies; since GHG accounting metrics used at present such as GWP do not provide accurate results for targets such as Net-Zero $CO_2$ due to the short life of some GHGs (Allen et al., 2018). To aid this use of GIR, we provide an Excel file containing the model with its default parameter set to give access to GIR for users unfamiliar with scientific programming languages. This version of the model could also be used to assist teaching of climate change and climate processes; and could even allow students access to an easy-to-understand model that they could use them-

selves to explore future scenarios and the relative impacts of future emissions of $CO_2$, $CH_4$ and $N_2O$; or the importance of climate sensitivity.

We aim to be able to provide gas-cycle parameter sets tuned to each of the CMIP6 ESMs in the near future, such that GIR is able to be used to emulate the full CMIP6 ensemble and to improve understanding of how these models differ in a single

consistent framework. We hope that such parameter sets may also provide users in other related fields such as climate policy who do not have experience with climate models access to a robust emulation of complex climate models with little learning curve.

*Code and data availability.* The model and code used to produce the figures is publicly available at https://github.com/njleach/GIR, and will be cleaned up and release ready prior to acceptance. All data used in this study is publicly available at the relevant cited sources.



*Author contributions.* NJL, SJ and MRA conceived the study. NJL and SJ wrote the model code, and BW helped tune model parameters.
JT provided CMIP6 response parameters. JL and MC advised on model uses and tested the model. NJL, CJS, ZN, JL and MRA wrote the
manuscript.

*Competing interests.* The authors declare that they have no competing interests.

*Acknowledgements.* We acknowledge the World Climate Research Programme, which, through its Working Group on Coupled Modelling,
coordinated and promoted both CMIP5 and CMIP6. We thank Jens Mühler (Scripps Institution of Oceanography) and Matthew Rigby
(University of Bristol) for providing up-to-date AGAGE inverse emissions data for halogenated greenhouse gases.







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
