# Peer review of "GIR v1.0.0: a generalised impulse-response model for climate uncertainty and future scenario exploration"

_Geoscientific Model Development, 2019_

## Referee Comment (RC1) · William Collins (Referee) · 25 Feb 2020

Leach et al. GIR

This paper proposes a set of equations to be used as a simple tool to compute temperate changes from given emission scenarios. This is certainly a worthwhile concept and this paper will make an important contribution to this goal. However, if the aim is to encourage wide uptake, considerable work is needed to make the paper more readable. The paper needs to explain the concepts more fully (one example out of many is the central role of iIRF100) but in a simpler way without assuming so much familiarity with the model.

My first point is rather minor, but it seems the justifications put forward for GIS are very similar to those originally put forward for FAIR. Did the authors find in developing FAIR that it became more complex than expected? Line 54 states "... representation of other greenhouse gases, significantly increasing the structural complexity of the FAIR model...". So is the main difference between GIS and FAIR the treatment of non-CO2 gases? Is this sufficiently different to be a new model?

My bigger criticism is on the "transparency" of the r0, ru, rT and ra coefficients. It is not at all clear that these translate readily into physically meaningful parameters that can be compared with and between models. This contrasts with the "Gas pools" and "Thermal boxes" that can be understood in terms of reservoirs and easily be compared eg with Joos et al. (2013) and Geoffroy et al. (2013). The formulation of alpha as a sinh is very non-intuitive, and so there is a concern that this will discourage the extent of the uptake of this model that the authors hope for. The defining principle in Millar et al. 2017 was the concept of the iIRF100. However, it was not obvious (or explained in Millar et al. ) why this should have been a fundamental quantity for CO2, and it isn't explained here why this should be a fundamental quantity for methane and N2O. Table 1 in Millar et al. 2017 gives guiding analogues for the r-terms, but it is not clear that they mean anything physical for methane or N2O. For instance the text refers to natural methane emissions being accounted for in the fit to get the r0 term, but it is not obvious there should be any connection between emissions and r0.

Related to the above, the fitting procedures are not clear, particularly the value in fitting to the historical observations. With 14 parameters in table 1, it is not surprising that the model can fit the historical record well, but is it for the right reasons? It would be more useful to fit to idealised experiments (as is done in Joos et al., and Geoffroy et al., and in the C4MIP experiments for beta and gamma). Then it should be clear which terms in the models are being represented by parameters in GIS.

The sections describing calculations of e.g. species lifetime "Emission-driven historical simulations", and climate responses metrics (TCR, TCRE) "Idealised experiments"

need to be clearer as to what extent these quantities are inputs to the models and to what extent new information is provided through the fits to historical timeseries. Similarly for the emission metrics such as GWP.

It would be useful to compare SSP3-70 results from FAIR and GIS since this scenario has very different levels of ozone precursors. This would confirm (or not) that the treatment of ozone forcing in GIS was sufficient.

Line 49: What was "not quite adequate" about AR5-IR? This needs to be more explicit.

Line 80: This section needs to start with some introduction and explanation of the concepts rather than immediately diving into the equations.

Line 85: Equation (3) is very non-intuitive, it is the solution to Eq 7 in Millar et al. 2017, but it seems to overcomplicate very uncertain relationships. While iIRF100 might have been a useful concept in Millar et al. 2017, it is not at all obvious that it is the most useful formulation for GIS. This is particularly true for methane and N2O since later on the equation needs to be linearised. Why not just leave it in a linear form? What is "h"? 100 years?

Line 93: In what way is the analytical equation an approximation, what terms have been neglected?

Line 116: Since GIS is representing carbon-cycle models, it would seem much more sensible to fit to emission-driven models, rather than bottom-up emissions to observations. If there is any discrepancy between emission-driven complex models and observed trends, then that represents a process we don't understand. Whereas in fitting to observations, any discrepancy will get folded into the fitted parameters in an unknown way and hidden. Can the rT and ru terms be related to the more physically relatable beta and gamma (either capital or lower case) of C4MIP?

Line 130: Presumably some fixed relationship between temperature change and water vapour change is used? This should be stated.

Line 131: As for CO2, this seems very dangerous. The suggestion from Smith et al. 2017 is that bottom-up emission estimates are not consistent with the observed concentrations. This fitting hides that by folding the inconsistencies into the fitted parameters. How many parameters are fitted? The sentence suggests the pre-industrial concentration is "specified" rather than fitted.

Line 146: What do these tuned parameters mean? In particular what does a r0 of 9.079 (years?) mean physically? The integral of a pulse of methane is equal to tau1=9.15 years, so it seems as if tau1 and r0 are degenerate. Similarly for N2O: r0 is just tau1(1-exp(-100/tau1)). Does r0 have any meaningful property that is different to tau1?

Table 1. These parameters all need units and guiding analogues.

Line 167 to 175: This explanation of the natural emissions needs expanding. Indeed in GIS, the natural emissions must be fixed at C0/r0 for methane and N2O. Smith et al. 2017 showed that the bottom-up emissions are inconsistent with the observed concentrations, so it is not clear how GIS can reproduce the historical concentrations from these emission (figure 2). How is Supplementary figure 2 generated if GIS can reproduce the observed concentration with constant natural emissions?

195: It is not clear whether the lifetimes presented here contain any new data, given the tau1 and ra are specified from the Prather and Holmes studies. Are the present and pre-industrial lifetimes just extrapolations based on those coefficients – if so, it is not surprising they agree. Again there is a suggested dependence on r0 - this one parameter seems to do a lot of work so there really does need to be a physical justification for it.

Line 220 -240: Again, this parameter fitting hides the science. The Etminan formulae are transparent, whereas the formulae in table 2 have different coefficients and additional terms. Are the non-primary coefficients significantly different from zero? The calculations for the f2 factors for the non-direct effects of methane on ozone and others need to be shown. I suggest sticking with the Etminan formulae and explicitly adding in

extra terms only when necessary to represent physical processes (CH4 - N2O overlap, ozone production). The calculations for these extra terms need to be provided.

Line 245: While ozone is historically correlated with CH4. The assumption of the same correlation continuing in future might not be valid. This could be quickly tested using the FAIR parameterisation for ozone.

Line 257: Where does this value of 60 MtSO2/yr come from?

Line 337: What is the full forcing for N2O? There is no additional forcing attributed to N2O in table 8.SM.6 of Myhre et al. 2013. The calculation used needs to be shown in full.

Line 339 "We find values comparable to the current literature": These calculations of IRF100, iIRF100, AGWP100 and GWP100 and their methodology do not add value to the paper if all that can be said is that they are comparable to current literature. I suggest this section is removed. The values for CO2 come from the a_i and tau_i parameters which come from Joos et al, so it is not surprising that these agree with Joos. Similarly for the methane and N2O metrics, these are determined by the tau_1 and the f_i parameters which are derived from the same Prather, Holmes and Etminan papers as used in the literature. It is possible that the added temperature dependence of some of these parameters could affect the metrics - if the authors think this is worthy of discussion then the difference between the metrics for variable alpha could be compared with alpha=1 values (which is implicitly what is assumed in the literature).

Lines 354-373: Again it is not clear that TCRE is a new result from GIS, rather than a consequence of the parameters adopted in GIS. The text seems to suggest that the TCRE agreement with literature is a validation of the model, whereas it seems mostly driven by the same inputs as the literature (Joos -like carbon response, and Geoffroy-like climate response).

Line 368-369 "...lowers the upper end of the TCRE distribution and raises the median."

If the authors are implying that using GIS can provide new information on these metrics, then this needs much more explanation on where this new information is coming from.

Line 380: The IIASA database has harmonised historical and future emissions, so it would make sense to use those rather than redoing this independently.

Line 388: $CO_2$, $N_2O$ and $CH_4$ concentrations are available on the IIASA SSP database (from MAGICC). Presumably the authors could quickly generate these from FAIR too.

Line 389: Why is the $CO_2$ concentration slightly lower than FAIR given that GIS and FAIR use the same formulation for $CO_2$?

Line 478: Given that the more complex chemistry and carbon-cycle models can't reproduce atmospheric concentrations from bottom-up emissions, it is extraordinary that a simple parameterisation of these complex models can do so.

Line 483: This GWP100 increase is not discussed earlier in the text. It shouldn't appear first in the conclusions. The value (35.3) is also different from any in table 4.

Line 484: The timing of peak warming is not discussed earlier in the text (apart from briefly for $CO_2$). This shouldn't form part of the conclusions unless it is discussed more fully earlier. – is it not just that $N_2O$ has a longer lifetime ($\sim$120 years) than the $\sim$10 years for methane and $\sim$4 and 36 years for $CO_2$, rather than any new finding from GIS?

---

## Referee Comment (RC2) · Anonymous Referee #2 · 26 Mar 2020

The manuscript "GIR v1.0.0: a generalised impulse-response model for climate uncertainty and future scenario exploration" present a simple climate model, that consists of six equations, that can easily be implemented in other models regardless of code language and also used in software such as Excel. The potential use of the model can therefore be widespread.

I have only minor comments to the manuscript as follows:

L 8. It could be highlighted more the tunable nature of the model. In other parts of the manuscript it is nicely written e.g. : "Here we emphasize that the models themselves are not systematically biased either low or high - it is the parameters used, and how

these are selected, that determines the model response. " This could be emphasized even more in the abstract.

L33. The differences between SCMs and ESMs here are runtime and lines of code. Complexity should also be mentioned. SCM: only global, global mean temperature, while ESMs three dimensional, gridded, large set of variables, not only temperature etc.

Figure 1: It could be useful if R, T, G, S, etc. could be defined in the figure caption.

Figure 2: Here, and elsewhere in the manuscript, references are written without a year. E.g: "from Meinshausen et al. concentrations". Add "(CMIP6 historical)" to the figure caption. Indicate end year in the figure. Or mention in the caption. I am not familiar with the use of TgN2O-N2. Better to use TgN instead? (same for L401)

Figure 3: Unit on the y-axis are missing.

Table 2: In the table caption, add which indirect forcing effects that are included in f2.

L245: Any reference to the high correlation for total ozone and CH4 concentration?

L 371: typo

L 385: "While this harmonization procedure may not exactly match the one used in the database scenarios" Do you mean the harmonization procedure that are used in the SSP database? A set of the scenarios are harmonized to historical emissions in the SSP database. Why did you not use them? And related to L387, isn't these harmonized emission scenarios in the SSP database what you ask for?

Also, regarding the scenarios, there are many ssp scenarios available, generated by different models. Please specify which ones you have used. I guess it is the unharmonized marker scenarios?

Figure 4, 5 and 6, there are more than hundred SSP scenarios. Replace a "range of SSP scenarios" with "three SSP scenarios"?

**GMDD**

Figure 4: Add in the figure caption why the dashed and dotted lines are not included in CH4 and N2O figure? Add what the shading and error bar in the figure represent.

Figure 5: In figure caption concentration is written. Replace by ERF. Also here indicate what the shading represents.

L 447: To fully emulate the CMIP6 ensamble, aerosol ERF must also be tuned. This is not mentioned in the text. Please discuss how this can be done.

In the conclusion section, first differences to FaIR v1.0 is presented, while later results are compared to FaiR v1.3. In the first part of conclusion, also present the differences to FaIR v1.3.

---

## Short Comment (SC1) · 31 Mar 2020

Just a minor comment on the citation of PRIMAP-hist:

For reproducibility, It would be good to reference the actual version used, as PRIMAP-hist HISTTP is only described in the data description of the 2.x downloads, see e.g.

http://doi.org/10.5880/PIK.2019.018

and

https://www.pik-potsdam.de/paris-reality-check/primap-hist/

---

## Referee Comment (RC3) · Anonymous Referee #3 · 3 Apr 2020

I read the manuscript with great interest, but I am afraid to say that the new model that the authors presented seems to me a combination of existing models, or an expansion or a generalization of the FaIR model. The novelties do not come very clear to me throughout the manuscript unfortunately. To begin with, the carbon cycle is essentially the same with the one in FaIR, except for some changes in feedback-related parameters (Table 1). The table indicates that the CH4 and N2O gas cycle representations in GIR are more complicated than in FaIR, but these are already considered by other SCMs like MAGICC. The forcing equations for three gases are either the Etminan parameterizations or their simplification without gas interactions. The climate model is the Tsutsui model (3-box) published before, in comparison to a 2-box model in FaIR.

The whole things above left me wonder how come the model deserves a new name. Is this a marketing strategy to sell the model again? In my eyes, the model appears like a re-tuned version of FaIR. It is not my intention to make it ironic, but the only reason to justify the new name can be to avoid using the name "FAIR" any more, which was previously used to call a simple climate model developed by a different group (den Elzen and Lucas 2005; den Elzen and van Vuuren 2007).

Now, from a different angle, I would think that the model would be an innovation if it is really simple and workable. But the current manuscript indicates that this does not seem to be the case. The authors claimed so by emphasizing that the model can be expressed just in six equations, so deserved a new name (Lines 59-60). The model appears simple at surface, but a closer examination easily reveals that the equations are aggregated at a general level, hiding the complexity. In fact, equation (3) is very complicated, and its physical interpretation is not obvious. I doubt that general users that the authors intend to reach out appreciate this equation.

At multiple places in the manuscript, the authors insist simplification, e.g. "our core aim of simplicity" in Line 221. If this is a guiding principle for this model, the simplification should be more strongly enforced and the model should be designed accordingly. But if I don't get it wrong, the current model is actually more complicated even than FaIR because there are more parameters and feedbacks for CO2, CH4, and N2O gas cycle in GIR and also because the climate model has now three boxes (two boxes in FaIR). The authors certainly separate the gas cycles "to simplify" by removing the interaction of CH4 and N2O forcing and the CH4-O3 interaction. But gas cycles are still indirectly linked through seemingly complicated temperature feedback in equation (3). If simplification is really a guiding principle, the authors need to embrace it more and think further what the minimum representation to adequately represent the global response of the earth system to greenhouse gas emissions is (in Line 100, authors refer to the Supplementary Information for such discussion, but I was not able to find it). This question has been asked by many simple climate modelers, but the answer might

be different now, given the latest knowledge and the current political situation after the adoption of the Paris Agreement. If the intended model use is limited to Paris-relevant low temperature stabilization pathways, certain feedbacks and model features may not be needed, which simplifies the model.

I have a general impression that the discussion in this paper is placed in a narrow range of papers. Many SCMs exist, but throughout the paper the authors do not really discuss SCMs other than FAIR and MAGICC. Where relevant, the paper should touch on other SCMs and their model features including but not limited to ACC2, BernSCM, CICERO SCM, Hector, OSCAR, and WASP. Also the SCM built in DICE should also be incorporated in the discussion. In my view, some innovation claimed by this paper (e.g. see my comment on L 48 to 60) is a result of the ignorance of other previous papers. The discussion needs to be widened in scope.

In summary, a substantial amount of work is required to revise the paper, potentially including further tuning or development of the model. My judgement is that this manuscript should be rejected, with an opportunity for resubmission. I provide further comments below. But the comments are not given comprehensively because I expect that the paper will be in a completely new form after revision. I am sorry that I cannot be positive in this review.

Further comments

L 36 to 46: The discussion in this paragraph seems to contradict with the statement in the abstract: "other methods would be equally valid." This also contradicts with the fact that MAGICC has been solely used in some previous IPCC WG3 Assessment Reports. The issue has been rather the dominant use of MAGICC, whose codes are not publicly available. The authors could push GIR to be used for assessments. But this should not be privileged to GIR. This should be open to other models complementary. I therefore disagree with the idea of one common SCM.

L 48 to 50: The model equation to calculate GHG metrics has been transparent in

previous IPCC Assessment Reports, to my knowledge. I disagree with the statement "that model was not quite adequate to reproduce the evolution of the integrated impulse response to emissions over time." See Joos et al. (2013).

L 48 to 60: It is unclear what "all of these innovations" are. Innovations need to be discussed in a wider context of previous studies. For example, the non-linearity of the carbon cycle has been introduced by Joos et al. (1996); Hooss et al. (2001).

L 85: I don't think that general users would understand this equation. This is explained in Lines 93-95 by citing Millar et al. (2017), but this needs elaboration.

L 100: I cannot find the discussion on the adequacy of this analytic form in Supplementary Information.

L 114-115: I cannot find the result that the authors refer to.

L 134 to 135: Many international assessments (e.g. CCAC) indicate that the CH4 and O3 interaction is very important for climate and clean air policies. If the model drops this interaction, this needs to be done more carefully with an extensive set of sensitivity analyses to find out what the limitations are. Many SCMs capture CH4-O3.

L 206: In Fig 2, the uncertainty range for N2O is not shown.

L 233 to 235: If this model is made public, some people would use it for RCP8.5 by forgetting (or ignoring) that the model is tuned only Paris-relevant scenarios. This tuning strategy may be risky.

Supplementary Information Table 1: Is this a common way to describe the unit for N2O?

References den Elzen MGJ, Lucas PL (2005) The fair model: A tool to analyse environmental and costs implications of regimes of future commitments. Environmental Modeling & Assessment 10:115-134

den Elzen MGJ, van Vuuren DP (2007) Peaking profiles for achieving long-term temperature targets with more likelihood at lower costs. Proceedings of the National Academy of Sciences 104:17931-17936

Hooss G, Voss R, Hasselmann K, Maier-Reimer E, Joos F (2001) A nonlinear impulse response model of the coupled carbon cycle-climate system (niccs). Clim Dyn 18:189-202

Joos F, Bruno M, Fink R, Siegenthaler U, Stocker TF, Le Quélé C, Sarmiento JL (1996) An efficient and accurate representation of complex oceanic and biospheric models of anthropogenic carbon uptake. Tellus B 48:397-417

Joos F, Roth R, Fuglestvedt JS, Peters GP, Enting IG, von Bloh W, Brovkin V, Burke EJ, Eby M, Edwards NR, Friedrich T, Frölicher TL, Halloran PR, Holden PB, Jones C, Kleinen T, Mackenzie FT, Matsumoto K, Meinshausen M, Plattner G-K, Reisinger A, Segschneider J, Shaffer G, Steinacher M, Strassmann K, Tanaka K, Timmermann A, Weaver AJ (2013) Carbon dioxide and climate impulse response functions for the computation of greenhouse gas metrics: A multi-model analysis. Atmospheric Chemistry and Physics 13:2793-2825

---

## Author Comment (AC1) · 2 Aug 2020

Response to William Collins, 02/08/2020

Our response is given in standard typeface, with the original review in italic.

*This paper proposes a set of equations to be used as a simple tool to compute temperate changes from given emission scenarios. This is certainly a worthwhile concept and this paper will make an important contribution to this goal. However, if the aim is to encourage wide uptake, considerable work is needed to make the paper more readable. The paper needs to explain the concepts more fully (one example out of many is the central role of iIRF100) but in a simpler way without assuming so much familiarity with the model.*

We thank the reviewer for their thorough and instructive review of the manuscript. In the following response, we address all the concerns raised on a point-by-point basis. In each case, we will either state what amendments are made to the revised manuscript following each comment, or explain any reason for disagreement.

*My first point is rather minor, but it seems the justifications put forward for GIS are very similar to those originally put forward for FAIR. Did the authors find in developing FAIR that it became more complex than expected? Line 54 states ". . . representation of other greenhouse gases, significantly increasing the structural complexity of the FAIR model. . .". So is the main difference between GIS and FAIR the treatment of non-CO2 gases? Is this sufficiently different to be a new model?*

We agree with this comment. This is largely resolved in our revision, since we now present an update to FaIR, rather than an entirely new model. In terms of the motivation for this update; we feel that while FaIR is an extremely useful and still relatively simple tool for exploring climate change, we believe that it is possible to simplify it even further without losing robustness and make it even more transparent with this update.

*My bigger criticism is on the "transparency" of the r0, ru, rT and ra coefficients. It is not at all clear that these translate readily into physically meaningful parameters that can be compared with and between models. This contrasts with the "Gas pools" and "Thermal boxes" that can be understood in terms of reservoirs and easily be compared eg with Joos et al. (2013) and Geoffroy et al. (2013). The formulation of alpha as a sinh is very non-intuitive, and so there is a concern that this will discourage the extent of the uptake of this model that the authors hope for. The defining principle in Millar et al. 2017 was the concept of the iIRF100. However, it was not obvious (or explained in Millar et al.) why this should have been a fundamental quantity for CO2, and it isn't explained here why this should be a fundamental quantity for methane and N2O. Table 1 in Millar et al. 2017 gives guiding analogues for the r-terms, but it is not clear that they mean anything physical for methane or N2O. For instance the text refers to natural methane emissions being accounted for in the fit to get the r0 term, but it is not obvious there should be any connection between emissions and r0.*

Some sentences better discussing the physical reasoning behind r parameters in CO2, CH4 and N2O formula have been added to the text.

The physical justification behind the alpha value in Millar et al's FaIRv1.0 is that it acts as a means to provide a state dependence to the timescales in the carbon cycle. The alpha value multiplied by the nominal time constants tau_1 <-> tau_4 produce real time carbon cycle timescales which are dependent on the carbon accumulated in the land, ocean and biosphere, and on the global average temperature anomaly.

The iIRF100 is the integrated airborne fraction over a 100 year period, and represents the extent to which a pulse emission of CO2 remains in the atmosphere over a 100 year period. We integrate the airborne fraction to get a parameter which distills the average airborne fraction over a 100 year period, as opposed to an instantaneous airborne fraction

at year 100. The iIRF100 can be exactly calculated with the FaIRv1.0 framework, and estimated using a parameterisation in terms of global average temperature and accumulated CO2 in the pools of the carbon cycle in order to estimate the value of the alpha parameter, i.e. how much has the current climate state impacted the carbon sink behaviours.

In FaIRv2.0 we have followed this framework but argued an impulse response structure is more generally applicable to describe the evolution of concentrations for a wide range of pollutants. The necessary step is to define adequate parameterisations such that the correct physics is captured for each gas. In the case of FaIRv1.0, the requirements were for CO2 a global atmospheric residence time for ~40% of input CO2 emissions of several centuries, and a carbon cycle feedback which depends on the accumulated carbon stock in the non-atmospheric pools of the carbon cycle (r_C) and the global average temperature anomaly (r_T).

For methane the requirements are different; we want a globally averaged atmospheric residence time of around a decade, and feedbacks which are dependent on the atmospheric concentration (r_a) and the global average temperature (r_T). Further there is only one major decay pathway for methane out of the atmosphere, and so one pool is sufficient (tau_2 <-> tau_4 = 0). This parameterisation is adequate, as is demonstrated in the paper text over historical period, to capture the globally averaged emissions to concentrations relationship for an SLCP such as methane.

For N2O the story is similar to CH4, except we require only a feedback which is dependent on the atmospheric concentration of N2O (r_a), again requiring a single pool (tau_2 <-> tau_4 = 0).

For minor contributing pollutants, such as HCFCs, we work with a simple parameterisation with a fixed lifetime (alpha*tau_1) which is representative of values quoted in the literature. This is analogous to the most simple exponential decay model.

As for the fit to get the r_0 requiring accounting for natural emissions of CH4/N2O: this is required because the r_0 parameter sets the baseline atmospheric decay lifetime for anthropogenic CH4/N2O. This lifetime is dependent on the background of natural emissions because the CH4/N2O lifetime is dependent on the atmospheric concentration of these gases.

*Related to the above, the fitting procedures are not clear, particularly the value in fitting to the historical observations. With 14 parameters in table 1, it is not surprising that the model can fit the historical record well, but is it for the right reasons? It would be more useful to fit to idealised experiments (as is done in Joos et al., and Geoffroy et al., and in the C4MIP experiments for beta and gamma). Then it should be clear which terms in the models are being represented by parameters in GIR.*

We will clarify the fitting procedures within the text and in accompanying publicly available code. Specifically addressing the 14 parameter fitting procedure: we do not fit to all 14 parameters simultaneously at any point (though this would presumably lead to an excellent fit). We fit groups of parameters in what we believe are physically motivated ways, basing our fits on either available data or literature. However, we entirely agree that the original submission was not clear enough with exactly how we fit each parameter, and we will expand on this section fully in the revision. In the revision, we have tried to avoid fitting parameters ourself except where necessary; instead taking values from the literature or using published methodologies.

*The sections describing calculations of e.g. species lifetime "Emission-driven historical simulations", and climate responses metrics (TCR, TCRE) "Idealised experiments" need to be clearer as to what extent these quantities are inputs to the models and to what extent new information is provided through the fits to historical timeseries. Similarly for the emission metrics such as GWP.*

We do not wish to suggest that the GIR metric values represent any advances in their estimation with this model study; since they are really just a function of the tuned parameters (and as stated other methodologies of parameter fitting would be equally valid to those chosen). We provided these values are they are useful for any consumers of the model who wish to use the default configuration without having to calculate these metrics themselves. As such, we have moved all the metric calculation sections of the paper to the supplementary material and made it clear that they are for model benchmarking purposes only.

*It would be useful to compare SSP3-70 results from FAIR and GIR since this scenario has very different levels of ozone precursors. This would confirm (or not) that the treatment of ozone forcing in GIR was sufficient.*

This is an extremely helpful suggestion. In the revision we include a comparison of FaIR v1.5 and GIR over the SSP3-70 scenario. We have made some adjustments to the ozone parameterisation presented, and a comparison with FaIR v1.5 and MAGICC7.0.1-alpha is now included.

*Line 49: What was "not quite adequate" about AR5-IR? This needs to be more explicit.*

This has been clarified in the text. While AR5-IR is a transparent, simple tool for calculating metrics at the present day- something it does extremely accurately, it cannot be used robustly to simulate either historical or future emission scenarios. This is due to the lack of a state dependence in the carbon cycle: the airborne fraction within AR5-IR remains constant throughout a simulation, so AR5-IR cannot emulate any changes in the efficacy of carbon sinks with rising atmospheric concentrations or temperatures (as we observe in Earth-system-models).

*Line 80: This section needs to start with some introduction and explanation of the concepts rather than immediately diving into the equations.*

Thank you for the advice, we have attempted to clarify and restructure the text to improve its readability.

*Line 85: Equation (3) is very non-intuitive, it is the solution to Eq 7 in Millar et al. 2017, but it seems to overcomplicate very uncertain relationships. While iIRF100 might have been a useful concept in Millar et al. 2017, it is not at all obvious that it is the most useful formulation for GIR. This is particularly true for methane and N2O since later on the equation needs to be linearised. Why not just leave it in a linear form? What is "h"? 100 years?*

The formulation of this equation is essentially directly inherited from the Millar et al paper. They proposed that the iIRF100 was a linear function of accumulation carbon stock in the land and ocean and temperature. Within the original FaIR framework, this linear dependence resulted in the state-dependence parameter, alpha, being found with a root finding routine (since alpha is a nonlinear function of the iIRF100, as equation 7 in Millar et al). However, this solution for alpha (for a very wide range of iIRF100 values) is very well approximated by the sum of a linear and exponential function. This is what results in the- admittedly rather menacing looking- equation 3.

We linearise the equation for N2O and CH4 only to enable a more transparent procedure for fitting the r coefficients for these gases, rather than just using an arbitrary optimisation routine. As these terms are small for realistic CH4/N2O concentrations, we believe this linearisation is defensible.

"h" is indeed 100 years- we have made this clear in the text.

*Line 93: In what way is the analytical equation an approximation, what terms have been neglected?*

The solution for alpha offered in Millar et al. (2017) requires solving for the root of a non-linear equation. The solution offered here is that of the form alpha = g0*sinh(iIRF100/g1). The two are not identical, and the latter is an approximation of the true non-linear solution for alpha over a wide range of iIRF100 values (though this non-linear solution is itself a subjective parameterisation and therefore which is more justifiable is debatable). The difference between the approximation (equation 3 in text) and the exact solution found in Millar et al. (2017) is shown in figure S1 of the supplementary material. The difference, although larger at very low and very high iIRF100 values, is small over the range of iIRF100 values encountered in real world scenarios, even in very high emissions scenarios such as RCP85.

*Line 116: Since GIR is representing carbon-cycle models, it would seem much more sensible to fit to emission-driven models, rather than bottom-up emissions to observations. If there is any discrepancy between emission-driven complex models and observed trends, then that represents a process we don't understand. Whereas in fitting to observations, any discrepancy will get folded into the fitted parameters in an unknown way and hidden. Can the rT and ru terms be related to the more physically relatable beta and gamma (either capital or lower case) of C4MIP?*

For clarity, we have altered the CO2 cycle tuning to match that in Jenkins et al (2018). We believe that fitting carbon-cycle parameters to observed emissions is a defensible way to select default parameters such that the model will reproduce historical observations; this has been done in previous model literature (Millar et al, 2017 and Smith et al, 2018 both fit to match historical emissions and concentrations). While we entirely agree that this could run the risk of folding up discrepancies into the parameters, such discrepancies likely exist in future scenarios that are simulated by GIR, and so we argue that it is better to have a model that by default can reproduce historical trends; rather than one that may represent our current understanding of processes involved but cannot reproduce historical trends. However, we emphasise that this is only one method for tuning the model, and others are equally valid. For example, one could tune GIR to reproduce the carbon cycles of individual CMIP6 models, something we aim to do in the future. We shall explore relating the our feedback terms to those used in C4MIP: this is a very helpful suggestion.

*Line 130: Presumably some fixed relationship between temperature change and water vapour change is used? This should be stated.*

We don't use a fixed relationship explicitly. Instead, we convert the best literature estimates of the CH4 lifetime sensitivity to both water vapour and tropospheric air (Holmes 2013) temperature to the GIR parameterised temperature feedback (r_T). In Holmes, these two variables are computed using fixed relationships with surface air temperatures; here we take those same relationships to compute the CH4 lifetime dependence on just those two, and fit the r_T coefficient to both simultaneously. We will clarify and state this in the text.

*Line 131: As for CO2, this seems very dangerous. The suggestion from Smith et al. 2017 is that bottom-up emission estimates are not consistent with the observed concentrations. This fitting hides that by folding the inconsistencies into the fitted parameters. How many parameters are fitted? The sentence suggests the pre-industrial concentration is "specified" rather than fitted.*

In line with our core aim that the fitting we do here is as minimal as possible, we chose (as in FaIR v1.0 and v1.3) to specify a pre-industrial value. In the revision, we have decided to fit only two parameters: r0 (analogous to the pre-industrial iIRF100) and rC, keeping the rT/rC ratio the same as in Millar et al (2017). This tuning procedure is identical to that in Jenkins et al (2018), which used a previous iteration of FaIR model. We will fully clarify the tuning procedures used in the revised text.

*Line 146: What do these tuned parameters mean? In particular what does a r0 of 9.079 (years?) mean physically? The integral of a pulse of methane is equal to tau1=9.15 years, so it seems as if tau1 and r0 are degenerate. Similarly for N2O: r0 is just tau1(1-*

*exp(-100/tau1)). Does r0 have any meaningful property that is different to tau1?*
*Table 1. These parameters all need units and guiding analogues.*

We have somewhat simplified the tuning procedure in the revision to aid clarity. The reviewer is correct that for single pool gases, r0 is essentially degenerate with tau1; and as such we specify r0 values such that alpha=1 for these gases at the start of the integration. We will provide guiding analogues and units in the revised text. To physical meaning behind r0 would be that it is the average airborne fraction over 100 years from a pulse emission.

*Line 167 to 175: This explanation of the natural emissions needs expanding. Indeed in GIR, the natural emissions must be fixed at C0/r0 for methane and N2O. Smith et al. 2017 showed that the bottom-up emissions are inconsistent with the observed concentrations, so it is not clear how GIR can reproduce the historical concentrations from these emissions (figure 2). How is Supplementary figure 2 generated if GIR can reproduce the observed concentration with constant natural emissions?*

Figure 2 shows the GIR response when the "best-estimate" bottom-up emission timeseries (from PRIMAP-HISTtp) are input. As other models have tended to benchmark their response to the RCP timeseries (eg. FaIRv1.5), the supplementary figure 2 shows the residual when the RCP emissions are subtracted from the GIR inversion of RCP concentrations. This figure therefore demonstrates the incompatibility (at least, without additional "natural" emissions as are included in FaIRv1.5) of the RCP database emission and concentration scenarios (which has been shown previously). Within the natural emissions section we aimed to demonstrate that GIR is not vastly different to other SCMs (since the emission residual is similar to the specified natural emissions in FaIR); and we suggest that the problems associated with these internally specified natural emission pathways (or similarly concentration driven runmode to the present day) require more attention- since, for example, they result in historical pathways being treated differently to future scenarios, possibly with a discontinuity at the present day. As present day trends are key in robust estimation of a wide range of policy-relevant quantities (one example being the carbon budget), we argue that it is worth at least exploring the idea of a a single consistent model over the full (historical and future) scenario, rather than changing model internals between the two. We will expand on this section and clarify the points we attempt to make.

*195: It is not clear whether the lifetimes presented here contain any new data, given the tau1 and ra are specified from the Prather and Holmes studies. Are the present and pre-industrial lifetimes just extrapolations based on those coefficients – if so, it is not surprising they agree. Again there is a suggested dependence on r0 - this one parameter seems to do a lot of work so there really does need to be a physical justification for it.*

Yes they are- the idea behind their inclusion is to demonstrate that our parameterisation of the Holmes/Prather studies has not significantly affected any of the relevant physical quantities. We will include more explanation of the r0 parameter.

*Line 220 -240: Again, this parameter fitting hides the science. The Etminan formulae are transparent, whereas the formulae in table 2 have different coefficients and additional terms. Are the non-primary coefficients significantly different from zero? The calculations for the f2 factors for the non-direct effects of methane on ozone and others need to be shown. I suggest sticking with the Etminan formulae and explicitly adding in extra terms only when necessary to represent physical processes (CH4 - N2O overlap, ozone production). The calculations for these extra terms need to be provided.*

We have re-parameterised the model in the revision, and will be explicit about the formulae used.

*Line 245: While ozone is historically correlated with CH4. The assumption of the same correlation continuing in future might not be valid. This could be quickly tested using*

*the FAIR parameterisation for ozone.*

We have slightly adjusted the ozone forcing calculation in line with a parameterisation from Ehhalt (2001) in the revision, which is much more similar to the parameterisation in FaIR. However, we explicitly compare the two in the supplement now.

*Line 257: Where does this value of 60 MtSO2/yr come from?*

It was from Stevens (2015). We have adjusted the aerosol cloud interaction scheme in the revision.

*Line 337: What is the full forcing for N2O? There is no additional forcing attributed to N2O in table 8.SM.6 of Myhre et al. 2013. The calculation used needs to be shown in Full.*

This arose from a misunderstanding of mine- we provided the "full" and "direct-only" based on the with/without cc feedback GWP values for N2O. Clearly this is not the same as direct and indirect forcing so the "full" value has been removed.

*Line 339 "We find values comparable to the current literature": These calculations of IRF100, iIRF100, AGWP100 and GWP100 and their methodology do not add value to the paper if all that can be said is that they are comparable to current literature. I suggest this section is removed. The values for CO2 come from the a_i and tau_i parameters which come from Joos et al, so it is not surprising that these agree with Joos. Similarly for the methane and N2O metrics, these are determined by the tau_1 and the f_i parameters which are derived from the same Prather, Holmes and Etminan papers as used in the literature. It is possible that the added temperature dependence of some of these parameters could affect the metrics - if the authors think this is worthy of discussion then the difference between the metrics for variable alpha could be compared with alpha=1 values (which is implicitly what is assumed in the literature).*

We agree that these do not add academic value to the paper. However, as they are useful for consumers of the model; and for model intercomparison, they are still included, but in the supplementary material.

*Lines 354-373: Again it is not clear that TCRE is a new result from GIR, rather than a consequence of the parameters adopted in GIR. The text seems to suggest that the TCRE agreement with literature is a validation of the model, whereas it seems mostly driven by the same inputs as the literature (Joos-like carbon response, and Geoffroy-like climate response).*

We are aware that some consumers (non-experts who want a reasonable physical model for their study) may use this paper when deciding which model to employ. We wanted to display a number of key results, which although unspectacular in terms of differences to other studies, demonstrate consistency with the other standard SCM choices i.e. that our simplifications have not come at the expense of performance in these key metrics

*Line 368-369 ". . .lowers the upper end of the TCRE distribution and raises the median." If the authors are implying that using GIR can provide new information on these metrics, then this needs much more explanation on where this new information is coming from.*

We will revisit this part of the text and revise / move to SI. These sections aren't aiming to introduce new information, but instead provide estimates of the key parameters to compare against other parameters.

*Line 380: The IIASA database has harmonised historical and future emissions, so it would make sense to use those rather than redoing this independently.*

We had problems with the IIASA database having a complete set of resources to sample from. We have shifted to the RCMIP database, which contains complete scenarios that

match the IIASA database and fill in the missing gaps, and can now provide this, and what is requested below.

*Line 388: CO2, N2O and CH4 concentrations are available on the IIASA SSP database (from MAGICC). Presumably the authors could quickly generate these from FAIR too.*

We had problems with the IIASA database having a complete set of resources to sample from. We have shifted to the RCMIP database of simple model scenarios instead, and can now provide this, and what is requested above.

*Line 389: Why is the CO2 concentration slightly lower than FAIR given that GIR and FAIR use the same formulation for CO2?*

While GIR is extremely similar to FaIR, the analytic approximation of alpha as function of the iIRF100 implemented in GIR results in very slightly different concentrations when compared to the original FaIR model, which solves for alpha at each timestep. On a more detailed level, it is because the pre-industrial iIRF100 value has to be slightly lower to give the same value of alpha in GIR when compared to FaIR; this lowered iIRF100 (and correspondingly, r_0) results in the marginally lower concentrations observed. The differences are still small when compared to the differences between different SCMs.

We use different parameters so expect different output. We have retuned the parameter set using updated datasets for emissions inputs and concentration outputs. FaIRv1.0 is an old tuning for CO2 (from 2017 datasets), so we update as part of the FaIRv2.0 release.

*Line 478: Given that the more complex chemistry and carbon-cycle models can't reproduce atmospheric concentrations from bottom-up emissions, it is extraordinary that a simple parameterisation of these complex models can do so.*

They certainly can't do it perfectly, but they can do it reasonably. We have some global understanding of the processes governing emissions inputs and concentrations outputs. We have to make an assumption as to the baseline pre-industrial concentration, but every simple model has to do this (or specify natural emissions), including FaIRv1.3

*Line 483: This GWP100 increase is not discussed earlier in the text. It shouldn't appear first in the conclusions. The value (35.3) is also different from any in table 4.*

Our apologies, this was erroneously left in from an earlier stage of the analysis. We have removed this value and the metric discussion from the conclusions, instead including metrics for the model in the SI.

*Line 484: The timing of peak warming is not discussed earlier in the text (apart from briefly for CO2). This shouldn't form part of the conclusions unless it is discussed more fully earlier. – is it not just that N2O has a longer lifetime (~120 years) than the ~10 years for methane and ~4 and 36 years for CO2, rather than any new finding from GIR?*

We will remove this from the conclusions and add to the supplementary information. In essence the differences are due to the different lifetimes of the gases combined with the thermal response timescales, but we thought that the result was interesting and potentially policy relevant. It is essentially the same computation as was done in Ricke (2014) for CO2 (peak warming occurs about a decade after emission), but applied to CH4 and N2O also, which we have not seen published before.

References
Millar, R. J., Nicholls, Z. R., Friedlingstein, P., & Allen, M. R. (2017). A modified impulse-response representation of the global near-surface air temperature and atmospheric concentration response to carbon dioxide emissions. Atmospheric Chemistry and Physics, 17(11), 7213–7228. https://doi.org/10.5194/acp-17-7213-2017

Holmes, C. D., Prather, M. J., Søvde, O. A., & Myhre, G. (2013). Future methane, hydroxyl, and their uncertainties: Key climate and emission parameters for future predictions. Atmospheric Chemistry and Physics, 13, 285–302. https://doi.org/10.5194/acp-13-285-2013

Jenkins, S., Millar, R. J., Leach, N., & Allen, M. R. (2018). Framing Climate Goals in Terms of Cumulative CO2-Forcing-Equivalent Emissions. Geophysical Research Letters. https://doi.org/10.1002/2017GL076173

---

## Author Comment (AC2) · 2 Aug 2020

Response to Reviewer 2, 02/08/2020

Our response is given in standard typeface, with the original review in italic.

*The manuscript "GIR v1.0.0: a generalised impulse-response model for climate uncertainty and future scenario exploration" present a simple climate model, that consists of six equations, that can easily be implemented in other models regardless of code language and also used in software such as Excel. The potential use of the model can therefore be widespread.*

We thank reviewer 2 for their careful review of the manuscript. We will attempt to address each comment below, our responses are written in blue. We will either agree and make a suitable change to the manuscript, or offer an explanation as to why we disagree with the suggestion.

*I have only minor comments to the manuscript as follows:*
*L 8. It could be highlighted more the tunable nature of the model. In other parts of the manuscript it is nicely written e.g. : "Here we emphasize that the models themselves are not systematically biased either low or high - it is the parameters used, and how these are selected, that determines the model response. " This could be emphasized even more in the abstract.*

Agreed. Abstract changed to include sentence discussing this point.

*L33. The differences between SCMs and ESMs here are runtime and lines of code. Complexity should also be mentioned. SCM: only global, global mean temperature, while ESMs three dimensional, gridded, large set of variables, not only temperature Etc.*

We have amended the text in line with this comment.

*Figure 1: It could be useful if R, T, G, S, etc. could be defined in the figure caption.*

We have amended the caption in line with this comment.

*Figure 2: Here, and elsewhere in the manuscript, references are written without a year. E.g: "from Meinshausen et al. concentrations". Add "(CMIP6 historical)" to the figure caption. Indicate end year in the figure. Or mention in the caption. I am not familiar with the use of TgN2O-N2. Better to use TgN instead? (same for L401)*

We have amended the text in line with this comment. We have altered to use the more familiar TgN.

*Figure 3: Unit on the y-axis are missing.*

We have amended the figure in line with this comment.

*Table 2: In the table caption, add which indirect forcing effects that are included in f2.*

We have amended the figure in line with this comment.

*L245: Any reference to the high correlation for total ozone and CH4 concentration?*

We have slightly adjusted the ozone forcing calculation in line with a parameterisation from Ehhalt (2001).

*L 371: typo*

We have corrected this.

*L 385: "While this harmonization procedure may not exactly match the one used in the database scenarios" Do you mean the harmonization procedure that are used in the SSP database? A set of the scenarios are harmonized to historical emissions in the SSP database. Why did you not use them? And related to L387, isn't these harmonized emission scenarios in the SSP database what you ask for?*
*Also, regarding the scenarios, there are many ssp scenarios available, generated by different models. Please specify which ones you have used. I guess it is the unharmonized marker scenarios?*

We have chosen to mitigate against this confusion over what data sources we use by using emission scenario data from the RCMIP dataset to simulate future scenarios with GIR. This data source has the advantage of being complete, in a single location, and has been used by other simple modelling groups such that we can be certain that their output is directly comparable to ours. We will be explicit about which specific scenarios we have used in the revised text.

*Figure 4, 5 and 6, there are more than hundred SSP scenarios. Replace a "range of SSP scenarios" with "three SSP scenarios"?*

Replaced in the text.

*Figure 4: Add in the figure caption why the dashed and dotted lines are not included in CH4 and N2O figure? Add what the shading and error bar in the figure represent.*

For the 4th figure, the dotted and dashed lines show the model derived concentration timeseries for each of CO2, CH4 and N2O. We haven't calculated CH4 and N2O concentration timeseries using MAGICC6 and FaIRv1.3 because they both require an estimate of background emissions from natural sources. We pulled MAGICC6 timeseries from the IIASA database of scenarios contributing to the IPCC SR15 report. Only CO2 concentrations are available here, and so this is what we plot. Thanks to the RCMIP effort, we will be able to provide comparisons for all variables for all models in the revision. Shading represents the central 5th-95th percentile range, and error bar shows the median and 5th-95th percentile range in 2100 (added to caption).

*Figure 5: In figure caption concentration is written. Replace by ERF. Also here indicate what the shading represents.*

We have replaced concentrations with ERF in the caption, and now indicate what the shading represents..

*L 447: To fully emulate the CMIP6 ensemble, aerosol ERF must also be tuned. This is not mentioned in the text. Please discuss how this can be done.*

We agree that this has not been discussed in enough detail, and will expand in the revised text.

*In the conclusion section, first differences to FaIR v1.0 is presented, while later results are compared to FaiR v1.3. In the first part of conclusion, also present the differences to FaIR v1.3.*

We agree with this comment. We chose to refer to FaIR v1.0 when discussing the difference in carbon cycle from FaIR, as while FaIR v1.3 introduced many new features, the carbon cycle had remained the same. However, we will ensure to make consistent comparisons throughout in the revised text.

---

## Author Comment (AC3) · 2 Aug 2020

Dear Robert,

Thanks very much for your comment, we will ensure that we reference the actual version used in the manuscript in the revision.
* * *

---

## Author Comment (AC4) · 2 Aug 2020

Response to Reviewer 2, 02/08/2020

Our response is given in standard typeface, with the original review in italic.

*I read the manuscript with great interest, but I am afraid to say that the new model that the authors presented seems to me a combination of existing models, or an expansion or a generalization of the FaIR model. The novelties do not come very clear to me throughout the manuscript unfortunately. To begin with, the carbon cycle is essentially the same with the one in FaIR, except for some changes in feedback-related parameters (Table 1). The table indicates that the CH4 and N2O gas cycle representations in GIR are more complicated than in FaIR, but these are already considered by other SCMs like MAGICC. The forcing equations for three gases are either the Etminan parameterizations or their simplification without gas interactions. The climate model is the Tsutsui model (3-box) published before, in comparison to a 2-box model in FaIR. The whole things above left me wonder how come the model deserves a new name. Is this a marketing strategy to sell the model again? In my eyes, the model appears like a re-tuned version of FaIR. It is not my intention to make it ironic, but the only reason to justify the new name can be to avoid using the name "FAIR" any more, which was previously used to call a simple climate model developed by a different group (den Elzen and Lucas 2005; den Elzen and van Vuuren 2007).*

*Now, from a different angle, I would think that the model would be an innovation if it is really simple and workable. But the current manuscript indicates that this does not seem to be the case. The authors claimed so by emphasizing that the model can be expressed just in six equations, so deserved a new name (Lines 59-60). The model appears simple at surface, but a closer examination easily reveals that the equations are aggregated at a general level, hiding the complexity. In fact, equation (3) is very complicated, and its physical interpretation is not obvious. I doubt that general users that the authors intend to reach out appreciate this equation.*

*At multiple places in the manuscript, the authors insist simplification, e.g. "our core aim of simplicity" in Line 221. If this is a guiding principle for this model, the simplification should be more strongly enforced and the model should be designed accordingly. But if I don't get it wrong, the current model is actually more complicated even than FaIR because there are more parameters and feedbacks for CO2, CH4, and N2O gas cycle in GIR and also because the climate model has now three boxes (two boxes in FaIR). The authors certainly separate the gas cycles "to simplify" by removing the interaction of CH4 and N2O forcing and the CH4-O3 interaction. But gas cycles are still indirectly linked through seemingly complicated temperature feedback in equation (3). If simplification is really a guiding principle, the authors need to embrace it more and think further what the minimum representation to adequately represent the global response of the earth system to greenhouse gas emissions is (in Line 100, authors refer to the Supplementary Information for such discussion, but I was not able to find it). This question has been asked by many simple climate modelers, but the answer might be different now, given the latest knowledge and the current political situation after the adoption of the Paris Agreement. If the intended model use is limited to Paris-relevant low temperature stabilization pathways, certain feedbacks and model features may not be needed, which simplifies the model.*

We respect the point that the reviewer is making here, and have a resolution for the revision: the model we present will be an update to the FaIR model, rather than a separate model. We believe that this reduces the confusion over the justification and implementation that GIR and FaIR shared. This update to FaIR, as discussed in our original submission, aims to make the model structurally simpler and more transparent, while still retaining the climate projection and emulation ability of the previous version.

While full specification of FaIR v1.5 and MAGICC7.0.1-alpha would involve a reasonably long list of equations detailing all of the parameterisations that differ by gas/aerosol species (see e.g. Appendix A of Meinshausen et al., 2011), here we provide a framework in

which you can parameterise all of the key components with a single equation set. It is true to say that if a reader understands the carbon cycle within the model, then they understand every gas cycle, since the feedback parameters and equations are identical. This identical treatment of all gases is what enables the model to a) run extremely quickly and b) be converted into almost any programming language, including excel; something that is not true of FaIR v1.5 or MAGICC7.0.1-alpha. It is this same structural simplicity that should allow anyone with a reasonable working knowledge of a programming language to code up their own version of the model, rather than relying on the code we have written ourselves; attempting to do this for MAGICC7.0.1-alpha or FaIR v1.5 would require a great deal more effort than it does for this model.

*I have a general impression that the discussion in this paper is placed in a narrow range of papers. Many SCMs exist, but throughout the paper the authors do not really discuss SCMs other than FAIR and MAGICC. Where relevant, the paper should touch on other SCMs and their model features including but not limited to ACC2, BernSCM, CICERO SCM, Hector, OSCAR, and WASP. Also the SCM built in DICE should also be incorporated in the discussion. In my view, some innovation claimed by this paper (e.g. see my comment on L 48 to 60) is a result of the ignorance of other previous papers. The discussion needs to be widened in scope.*

There are papers/projects in press (such as the Reduced Complexity Model Intercomparison Project, RCMIP) which attempt to do a thorough comparison between the full range of simple climate models; this is not our aim with this paper, and we feel it would end up unnecessarily lengthening the paper. Our aim is two-fold: 1) to provide a simple climate model which is straightforward to implement in a wide range of settings, and simple to understand due to the minimal equation set, and 2) to produce and emulator which can behave to within a reasonable approximation of any other model (SCMs, MICs and GCMs).

We argue the model presented fills a gap not adequately filled by the SCMs available today. Many of the models mentioned are significantly more complex than FaIR v2.0. While several of them simulate features that cannot be simulated with FaIR v2.0 (such as computing carbon fluxes into specific sinks), they can also not be written down in just 6 equations and coded up in just a few hours. This is the gap that we feel FaIR v2.0 occupies, something we shall try to make clear in the revision.

*In summary, a substantial amount of work is required to revise the paper, potentially including further tuning or development of the model. My judgement is that this manuscript should be rejected, with an opportunity for resubmission. I provide further comments below. But the comments are not given comprehensively because I expect that the paper will be in a completely new form after revision. I am sorry that I cannot be positive in this review.*

Our revised paper is significantly different and improved compared to our original submission. This review was extremely informative in terms of guiding what we could improve upon in the revision.

*Further comments*
*L 36 to 46: The discussion in this paragraph seems to contradict with the statement in the abstract: "other methods would be equally valid." This also contradicts with the fact that MAGICC has been solely used in some previous IPCC WG3 Assessment Reports. The issue has been rather the dominant use of MAGICC, whose codes are not publicly available. The authors could push GIR to be used for assessments. But this should not be privileged to GIR. This should be open to other models complementary. I therefore disagree with the idea of one common SCM.*

The statement in the abstract refers to the default GIR parameterisation procedure rather than the model used.

It has been argued that there are fundamental differences in the response of individual SCMs (Schwarber et al 2019), which are in reality largely the result of inconsistent parameterisation. For example, there seems to be a widely held belief that FaIR is "cooler" than MAGICC7.0.1-alpha. Some of these authors admit that this belief is due in part to their previous work (see Leach, et al, 2018); but the more recent literature referenced above has not helped. This is not true - it may be that the choices made by the modelling groups in parameterising the models have led to one model running cooler than the other, but the models themselves are not "hotter" or "cooler". We hope to make this point clear throughout the text.

This paper argues that releasing a model with only a single 'standard parameter set' is misleading because it implies the model results are intrinsic to the characteristics of the model equation set, and not down to the parameter set chosen. We therefore wish to be clear in the text that although we have chosen to tune the model parameters in a particular way, this choice is not the only possible option. This is what we meant by "other methods would be equally valid"; for example one could tune to the model output of a particular ESM rather than observations, or to ESM multi-model mean output.

We aim to demonstrate that this model is able to convey the full range of climate responses seen in complex models (CMIP6) and observations through parameterisation of six equations. Being able to write down these equations and the parameters used could make results between chapters/working groups of the IPCC process much more coherent, even if they then chose to use different models for more thorough analysis or to study particular feature of the climate system not included in this model. We are not advocating for the cessation of all other SCM research.

*L 48 to 50: The model equation to calculate GHG metrics has been transparent in previous IPCC Assessment Reports, to my knowledge. I disagree with the statement*
*"that model was not quite adequate to reproduce the evolution of the integrated impulse response to emissions over time." See Joos et al. (2013).*

The model equation in, for example AR5 is indeed an extremely simple and transparent tool for this purpose of calculating GHG metrics for the present-day state of the climate. However, as demonstrated in Millar (2017), the IPCC AR5 Impulse-Response model does not capture the full response to pulse emissions of $CO_2$ observed in ESMs, failing to reproduce future and historical emission -> concentration pathways well in comparison to eg. MAGICC7.0.1-alpha6 or more complex ESMs due to the constant airborne fraction implied by the model; this deficiency was overcome by Millar et al through the introduction of a state dependence.

*L 48 to 60: It is unclear what "all of these innovations" are. Innovations need to be discussed in a wider context of previous studies. For example, the non-linearity of the carbon cycle has been introduced by Joos et al. (1996); Hooss et al. (2001).*

The 'innovations' are stepping the reader through a timeline of developments in the FaIR model code, from first inception in the AR5 chapter 8 supplementary material, through a carbon-only model in Millar et al. 2017, to a full SCM encompassing the full range of GHGs in Smith et al. 2018. We felt that the wider context regarding Joos work on the non-linearity of the carbon cycle or Meinhausens work on complete SCMs working on the full range of GHGs like MAGICC was supplementary to the aim of this text, but in the revision some further context has been added.

*L 85: I don't think that general users would understand this equation. This is explained in Lines 93-95 by citing Millar et al. (2017), but this needs elaboration.*

We shall expand our discussion of this equation and the relevant parameters in the revision, while still avoiding excessive repetition of the discussion already in Millar et al (2017).

*L 100: I cannot find the discussion on the adequacy of this analytic form in Supplementary Information.*

We have edited the text to include some clarification for the chosen analytic approximation of the form for alpha in this case in the text. The figure the text refers to in the SI is figure 1, which shows the computed alpha value for FaIRv1.3 and for the updated version presented. They agree closely over a large range of iIRF100 values. The caption gives further insight into differences at low and high iIRF100 values.

*L 114-115: I cannot find the result that the authors refer to.*

This result refers to figure 2, where historical concentration timeseries are computed given PRIMAP hist emissions inputs. We have edited text to explicitly refer to figure 2 here.

*L 134 to 135: Many international assessments (e.g. CCAC) indicate that the CH4 and O3 interaction is very important for climate and clean air policies. If the model drops this interaction, this needs to be done more carefully with an extensive set of sensitivity analyses to find out what the limitations are. Many SCMs capture CH4-O3.*

We will provide a comparison of the CH4 lifetime in FaIR v2.0 and Holmes (2013) in the supplement for reference. We note that the previous iteration of the FaIR model did not include any parameterisations of CH4 atmospheric chemistry.

*L 206: In Fig 2, the uncertainty range for N2O is not shown.*

We have revised the way in which uncertainties are computed in FaIRv2.0, retaining the behaviour of the previous iteration: uncertainties (with the exception of the carbon cycle) are introduced at the forcing step for probabilistic simulations. This comment is therefore no longer relevant.

*L 233 to 235: If this model is made public, some people would use it for RCP8.5 by forgetting (or ignoring) that the model is tuned only Paris-relevant scenarios. This tuning strategy may be risky.*

We disagree that we have explicitly tuned for only Paris-relevant scenarios; in general we have tended to err on the side of tuning to observations. Even though the errors relative to the Etminan formulae are larger for the higher-emission scenarios, the maximum error is very small compared to the total forcings observed in these emission scenarios. Relative to the many other uncertainties involved in the simulation of these worst-case scenarios, we suggest that this error is acceptable, given the caveat provided in the text. It is worth noting that the line-by-line forcing calculation in Etminan has an associated error of 10%, considerably larger than the error introduced through our tuning procedure. We will make sure to fully compare our forcing tunings to the Etminan OLBL data in the revision in a clear manner to demonstrate the potential cases in which our parameterisation breaks down.

*Supplementary Information Table 1: Is this a common way to describe the unit for N2O?*

The most common unit used for N2O emissions is usually expressed as TgN. However, this appears in most cases to mean Tg of nitrogen (N2). We decided to be explicit about this in the original submission, however we will use the standard nomenclature in the revision.

References
Nicholls, Z. R. J., Meinshausen, M., Lewis, J., Gieseke, R., Dommenget, D., Dorheim, K., … Xie, Z. (2020). Reduced complexity model intercomparison project phase 1: Protocol, results and initial observations. Geoscientific Model Development Discussions, 1–33. https://doi.org/10.5194/gmd-2019-375

Meinshausen, M., Raper, S. C. B., & Wigley, T. M. L. (2011). Emulating coupled atmosphere-ocean and carbon cycle models with a simpler model, MAGICC6 – Part 1: Model description and calibration. Atmospheric Chemistry and Physics, 11(4), 1417–1456. https://doi.org/10.5194/acp-11-1417-2011

Leach, N. J., Millar, R. J., Haustein, K., Jenkins, S., Graham, E., & Allen, M. R. (2018). Current level and rate of warming determine emissions budgets under ambitious mitigation. Nature Geoscience, 11(8), 574–579. https://doi.org/10.1038/s41561-018-0156-y

Schwarber, A. K., Smith, S. J., Hartin, C. A., Aaron Vega-Westhoff, B., & Sriver, R. (2019). Evaluating climate emulation: Fundamental impulse testing of simple climate models. Earth System Dynamics, 10(4), 729–739. https://doi.org/10.5194/esd-10-729-2019

Etminan, M., Myhre, G., Highwood, E. J., & Shine, K. P. (2016). Radiative forcing of carbon dioxide, methane, and nitrous oxide: A significant revision of the methane radiative forcing. Geophysical Research Letters, 43(24), 12,614-12,623. https://doi.org/10.1002/2016GL071930

Millar, R. J., Nicholls, Z. R., Friedlingstein, P., & Allen, M. R. (2017). A modified impulse-response representation of the global near-surface air temperature and atmospheric concentration response to carbon dioxide emissions. Atmospheric Chemistry and Physics, 17(11), 7213–7228. https://doi.org/10.5194/acp-17-7213-2017

Holmes, C. D., Prather, M. J., Søvde, O. A., & Myhre, G. (2013). Future methane, hydroxyl, and their uncertainties: Key climate and emission parameters for future predictions. Atmospheric Chemistry and Physics, 13, 285–302. https://doi.org/10.5194/acp-13-285-2013

*References*
*den Elzen MGJ, Lucas PL (2005) The fair model: A tool to analyse environmental and costs implications of regimes of future commitments. Environmental Modeling & Assessment 10:115-134*
*den Elzen MGJ, van Vuuren DP (2007) Peaking profiles for achieving long-term temperature targets with more likelihood at lower costs. Proceedings of the National Academy of Sciences 104:17931-17936*
*Hooss G, Voss R, Hasselmann K, Maier-Reimer E, Joos F (2001) A nonlinear impulse response model of the coupled carbon cycle-climate system (niccs). Clim Dyn 18:189-202*
*Joos F, Bruno M, Fink R, Siegenthaler U, Stocker TF, Le Quélé C, Sarmiento JL (1996) An efficient and accurate representation of complex oceanic and biospheric models of anthropogenic carbon uptake. Tellus B 48:397-417*
*Joos F, Roth R, Fuglestvedt JS, Peters GP, Enting IG, von Bloh W, Brovkin V, Burke EJ, Eby M, Edwards NR, Friedrich T, Frölicher TL, Halloran PR, Holden PB, Jones C, Kleinen T, Mackenzie FT, Matsumoto K, Meinshausen M, Plattner G-K, Reisinger A, Segschneider J, Shaffer G, Steinacher M, Strassmann K, Tanaka K, Timmermann A, Weaver AJ (2013) Carbon dioxide and climate impulse response functions for the computation of greenhouse gas metrics: A multi-model analysis. Atmospheric Chemistry and Physics 13:2793-2825*